# The landscape of regulatory genes in brain-wide neuronal phenotypes of a vertebrate brain

Hui Zhang[1,2,3†], Haifang Wang[1,3†], Xiaoyu Shen[1,3†], Xinling Jia[1,3], Shuguang Yu[1,2,3], Xiaoying Qiu[1,3], Yufan Wang[1,2,3], Jiulin Du[1,3,4*], Jun Yan[1,3,4*], Jie He[1,3*]

[1]Institute of Neuroscience, State Key Laboratory of Neuroscience, Center for Excellence in Brain Science and Intelligence Technology, Chinese Academy of Sciences, Shanghai, China; [2]University of Chinese Academy of Sciences, Beijing, China; [3]Shanghai Center for Brain Science and Brain-Inspired Intelligence Technology, Shanghai, China; [4]School of Future Technology, University of Chinese Academy of Sciences, Beijing, China

**\*For correspondence:**
forestdu@ion.ac.cn (JD);
junyan@ion.ac.cn (JY);
jiehe@ion.ac.cn (JH)

[†]These authors contributed equally to this work

**Competing interest:** The authors declare that no competing interests exist.

**Abstract** Multidimensional landscapes of regulatory genes in neuronal phenotypes at whole-brain levels in the vertebrate remain elusive. We generated single-cell transcriptomes of ~67,000 region- and neurotransmitter/neuromodulator-identifiable cells from larval zebrafish brains. Hierarchical clustering based on effector gene profiles ('terminal features') distinguished major brain cell types. Sister clusters at hierarchical termini displayed similar terminal features. It was further verified by a population-level statistical method. Intriguingly, glutamatergic/GABAergic sister clusters mostly expressed distinct transcription factor (TF) profiles ('convergent pattern'), whereas neuromodulator-type sister clusters predominantly expressed the same TF profiles ('matched pattern'). Interestingly, glutamatergic/GABAergic clusters with similar TF profiles could also display different terminal features ('divergent pattern'). It led us to identify a library of RNA-binding proteins that differentially marked divergent pair clusters, suggesting the post-transcriptional regulation of neuron diversification. Thus, our findings reveal multidimensional landscapes of transcriptional and post-transcriptional regulators in whole-brain neuronal phenotypes in the zebrafish brain.

## Introduction

The vertebrate brain harbors highly diverse neuronal types that are specifically interconnected to form functional circuits (*Kepecs and Fishell, 2014*; *Armañanzas and Ascoli, 2015*; *Moffitt et al., 2018*). The brain comprises conserved major cell types (neurons, progenitors, glia, endothelial cells, etc.) (*Marques et al., 2016*; *Saunders et al., 2018*; *Hodge et al., 2019*). Previous studies have indicated that neuronal types could be well characterized by their features, including electrophysiological properties, neurotransmitter/modulator identity, synaptic connectivity, brain region identity, and cellular morphology (*Zeisel et al., 2015*; *Pandey et al., 2018*; *Zeisel et al., 2018*). How these neuronal features are molecularly encoded at the whole-brain level is an important issue in neuroscience.

With the advent of single-cell RNA-sequencing (scRNA-seq) technology, recent studies have begun to characterize neuronal diversity by analyzing single-cell transcriptomes of large populations in the whole brain or specific brain regions of mice and humans (*Darmanis et al., 2015*; *Lake et al., 2016*; *Poulin et al., 2016*; *Tasic et al., 2016*; *Chen et al., 2017*; *Tasic et al., 2018*). These studies have provided compelling evidence that individual neuronal types could be well characterized by their transcriptomes (*Poulin et al., 2016*; *Tasic et al., 2018*; *Sugino et al., 2019*). For instance, scRNA-seq analyses of half a million cells from the mouse brain showed that neurons could be classified by genes

**eLife digest** The brain harbors an astounding diversity of interconnected cells. Each cell contains the same basic set of genetic instructions, but only a fraction of the genome is used in each cell to assemble proteins. This selective gene expression gives rise to each cell's characteristic properties, such as their shape and location, or whether they can activate or inhibit neighbouring cells.

How these defining features are encoded on a genetic level and selectively activated in cells to produce such diversity in the brain is not fully understood. One way to study gene expression in single cells involves profiling the transcriptome, the full range of intermediary RNA molecules a cell produces from its genes to make proteins.

Zhang et al. used transcriptome profiling to better understand how thousands of regulatory genes encoding regulatory proteins called transcription factors create different types of neurons in the zebrafish brain, which is similar to but much simpler than the human brain. To do so, they analysed transcriptome data extracted from cell populations located in specific brain regions and displaying different properties.

Zhang et al. identified distinct clusters of neurons in the larval zebrafish brain. Mathematical models then analysed the transcriptome profiles of these neuronal clusters with characteristic features. They revealed that neurons with similar characteristics did not necessarily share the same transcription factors. In other words, distinct sets of transcription factors gave rise to the same types of cells. Zhang et al. described this observation as a 'convergent' pattern. On the contrary, some neurons with dissimilar features expressed the same sorts of transcription factors, suggesting a 'divergent' developmental pattern also exists.

In summary, this work sheds light on variable gene expression patterns akin to design principles that shape neuronal diversity in the brain. It gives a new appreciation of how neuronal subtypes result from a complex set of regulatory factors controlling gene expression.

related to neuronal connectivity, synaptic transmission, and membrane conductance (*Zeisel et al., 2018*). Besides, the scRNA-seq analysis of nearly 200 genetically marked mouse neuronal populations also showed that neurons could be classified by the expression level of various transcription factors (TFs), ion channels, synaptic proteins, and cell adhesion molecules (*Sugino et al., 2019*). These studies provided extensive information on the molecules that could be used to define neuronal types.

Generally, these molecules could be sub-divided into two primary categories: regulatory genes and effector genes. Regulatory genes encode proteins involved in gene transcription and translation (e.g., TFs and post-transcriptional regulators), while effector genes encode proteins serving specific neuronal terminal features (e.g., synaptic proteins, ion channels, transporters, and receptors) (*Kratsios et al., 2015*; *Zeisel et al., 2015*; *Paul et al., 2017*, *Zeisel et al., 2018*; *Reilly et al., 2020*). Regulatory genes are critical for establishing and maintaining effector gene profiles that resulted in diverse neuronal types. Interestingly, distinct TFs can determine neuronal types with similar neurotransmitter identities in the *Caenorhabditis elegans*, arguing for phenotypic convergence (*Serrano-Saiz et al., 2013*; *Gendrel et al., 2016*; *Hobert and Kratsios, 2019*). Remarkably, this phenotypic convergence has also been reported in the *Drosophila*'s optic lobe (*Konstantinides et al., 2018*). However, the multidimensional landscapes of regulatory genes in vertebrate whole-brain neuronal phenotypes remain elusive.

The larval zebrafish brain comprises only about 100,000 cells, thus providing an outstanding vertebrate model for studying cell diversity within the entire brain, using single-cell transcriptome analysis with full cell coverage by the 10× Genomics Platform. Previous scRNA analysis of the zebrafish nervous system has elegantly demonstrated the temporal dynamics of brain cell development (*Raj et al., 2018*; *Raj et al., 2020*). In this study, we generated the multidimensional landscape of regulator genes in effector gene-based neuronal phenotypes (terminal features) at the whole-brain level by combining single-cell transcriptome data obtained from the whole brain, specific brain regions, as well as neurotransmitter- and neuromodulator-defined neuronal populations. We found that, at the transcriptional and post-transcriptional levels, glutamatergic/GABAergic neurons with the same terminal features could express different TF profiles, while those with different terminal features could express the same TF but different RNA-binding protein (RBP) profiles. In contrast, neuromodulator-type

neurons that display particular terminal features expressed unique TF profiles. Thus, our findings reveal multidimensional landscapes of transcriptional and post-transcriptional regulators in the whole zebrafish brain.

## Results

### Molecular classification of whole-brain cells in zebrafish

To uncover the transcriptomic profiles of diverse cell types with regional identity in the larval zebrafish whole brain at single-cell resolution, we dissected and dissociated cells from the whole brain (n = 4), four specific brain regions (n = 2 each), including the forebrain (Fore), optic tectum (OT), hindbrain (Hind), and the region underneath the optic tectum (sub-OT) in the 8 days post fertilization (dpf) zebrafish (*Figure 1—figure supplement 1A*). We performed scRNA-seq of these cells using the 10× Genomics Chromium 3′ v2 platform. The libraries were sequenced to a mean depth of 126,651 reads per library, with a median of 1891 UMI and 866 genes per cell (*Figure 1—source data 1*). Reproducibility of transcription analysis was shown by the finding that replicates of whole-brain samples and individual brain regions were primarily overlapped in the t-distributed stochastic neighbor embedding (t-SNE) plots (*Figure 1—figure supplement 1B-C*). We obtained the transcriptomes of 65,253 cells and selected 45,746 cells for further analysis after filtering out low-quality data (Materials and methods). Then aggregated all cells into 68 clusters using high-variance genes (n = 1402) in t-SNE plot (*Figure 1A*). The Jaccard index-based analysis showed that most clusters had a high Jaccard index greater than 0.6, indicating the robustness of these clusters (details in Materials and methods) (*Tang et al., 2020*) except for a few neuronal clusters from sub-OT (Clusters 52, 66), and non-neuronal clusters (neuroprogenitors: Clusters 37, 56; radial astrocytes: 65, *Figure 1—figure supplement 1E*).

Each cluster was annotated according to cell-type-specific marker genes from the literature or ZFIN (Zebrafish Information Network) database (*Figure 1—source data 2*). To assign the brain region identity for each cluster, cells from each of the four specific regions (Fore, OT, sub-OT, Hind) were found to cover multiple but non-overlapping clusters and all 68 clusters could be assigned with their brain region origins (*Figure 1—figure supplement 2A-B*, and *Figure 1—source data 3*). Furthermore, to identify potential brain region-specific markers that exist in all cell types, we identified all genes that were differentially expressed in each region for all major cell types ($vglut^+$, glutamatergic neurons; $gad^+$, inhibitory neurons; $pcna^+$, neuroprogenitors; $cx43^+$, radial astrocytes). The differentially expressed genes shared by all cell types were considered to represent the targeted region-specific markers (*Figure 1C*). We indeed identified a small set of genes specific to each brain region independent of cell type. For instance, *foxg1a*, *en2a*, and *hoxb3a* were explicitly expressed in all cell types of the Fore, OT, and Hind, respectively (*Figure 1D*). Interestingly, these brain region-specific genes also exhibited a conserved region-specific expression pattern in the mouse brain (*Hanks et al., 1995*, *Manzanares et al., 2001*, *Kumamoto and Hanashima, 2017*). We found no specific gene for the sub-OT, probably due to the diverse brain structures in this region. These region-specific genes, which may be involved in forming regional identity during brain development, could be used to study region-specific neuronal connectivity and function.

To assign the neurotransmitter/modulator identity for each cluster, we used the marker genes that were specific to primary neurotransmitter/modulator phenotypes, including *slc17a6b* (glutamatergic), *gad1b* (GABAergic), *slc6a5* (glycinergic), *th* (dopaminergic [DA]), *tph2* (serotonergic [5-HT]), and *chata* (cholinergic [ChAT]) (*Figure 1—figure supplement 2C*, *Figure 1—source data 3*). The ratio of glutamatergic to GABAergic neurons was the highest in the forebrain and lowest in the hindbrain, indicating that glutamatergic neurons predominantly belonged to the forebrain, whereas glycinergic neurons mainly resided in the hindbrain (*Figure 1—figure supplement 2D*). These regional patterns of neurons expressing different neurotransmitter types were validated using the transgenic fishlines: Tg (*vglut2a*:loxp-DsRed-loxp-GFP), Tg (*gad1b*:EGFP), and Tg (*glyT2*:GFP), each exhibiting distinct labeling of glutamatergic, GABAergic, and glycinergic neurons, respectively (*Figure 1—figure supplement 2E*).

Moreover, the Lawson-Hanson algorithm for non-negative least squares (NNLS) analysis using cluster-specific marker genes (top 20, *Figure 1—source data 4*) showed that these 68 clusters exhibited a high overlap with their counterparts in the juvenile zebrafish brain recently reported (*Raj et al., 2018*). Meanwhile, clusters with different regional origins or cell types also exhibited a high correlation

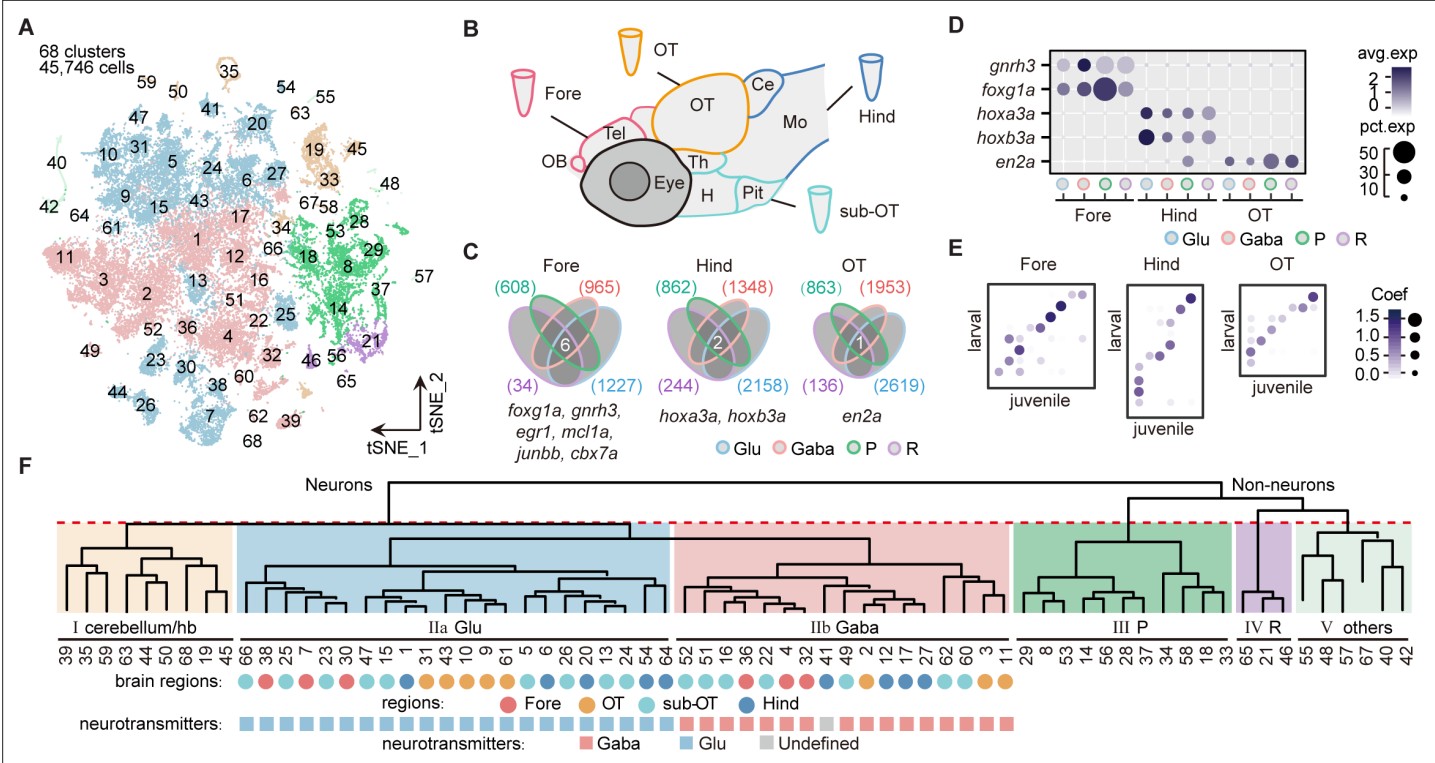

**Figure 1.** Molecular classification of whole-brain cells in larval zebrafish. (**A**) The t-distributed stochastic neighbor embedding (t-SNE) plot of 45,746 single-cell transcriptomes pooled from whole brains (n = 4) and four different individual brain regions (n = 2 each). The pooled cells were aggregated into 68 clusters, marked by a number. Each color-coded the major cell type as F. (**B**) The schematic showing different samples separately examined by single-cell RNA-sequencing on 10× Genomics Drop-seq platform: whole brain (n = 4), forebrain (Fore, n = 2), optic tectum (OT, n = 2), hindbrain (Hind, n = 2), and the region underneath the optic tectum (sub-OT, n = 2). OB: olfactory bulb; Tel: telencephalon; OT: optic tectum; Th: thalamus; H: hypothalamus; Pit: pituitary; Ce: cerebellum; MO: medulla oblongata. (**C**) Venn plots showing the differentially expressed genes in four major cell types identified by cell-type marker genes ($vglut^+$, glutamatergic neurons, Glu; $gad1b^+$, GABAergic neurons, Gaba; $pcna^+$, neuroprogenitors, P; $cx43^+$, radial astrocytes, R) in three brain regions (Fore, Hind, and OT). Commonly expressed genes in all cell types for a given brain region were identified as region-specific genes: six for forebrain (Fore), one for optic tectum (OT), and one for hindbrain (Hind), with genes listed below. (**D**) Dot plot showing the expression levels of region-specific marker genes in four major cell types (colored circles as C) in three brain regions. The gray level represents the average expression; dot size represents the percentage of cells expressing the marker genes. (**E**) Lawson-Hanson algorithm for non-negative least squares (NNLS) analysis showed cell clusters of Fore, OT, and Hind exhibited a high correlation with their counterparts of the juvenile zebrafish. Degree of correlation in marker genes is coded by the gray level and size of circle. (**F**) The dendrogram for the taxonomy of 68 identified clusters based on effector gene profiles (n = 1099). Main branches of neuronal and non-neuronal cells were classified into six branches (red dashed line) that include: I, cerebellum and habenula (hb); IIa, glutamatergic neurons (Glu); IIb, inhibitory neurons (Gaba); III, neuroprogenitors (**P**); IV, radial astrocytes (**R**); V, others, including microglia, endothelial cells, and oligodendrocytes. The colored dots and squares below indicate their regional origins and neurotransmitter-type, respectively.

The online version of this article includes the following source data and figure supplement(s) for figure 1:

**Source data 1.** Bioinformatics processing of raw reads of single-cell samples.

**Source data 2.** The annotation of 68 clusters of whole-brain sample.

**Source data 3.** The regional origins and neurotransmitter-type annotation of each whole-brain cluster with well-known markers.

**Source data 4.** Top 20 marker genes of whole-brain larval zebrafish 68 clusters.

**Source data 5.** Marker genes of major six cell type in whole brain.

**Figure supplement 1.** Molecular classification of whole-brain cells in larval zebrafish brain.

**Figure supplement 2.** Molecular classification of whole-brain cells in larval zebrafish brain.

with their counterparts in the juvenile zebrafish (*Figure 1E* and *Figure 1—figure supplement 2F*). Thus, our analysis indicated that the brain at 8 dpf mostly represented cellular diversity in the juvenile brain.

Furthermore, Gene Ontology (GO) analysis of 1402 variable genes used for the classification of all 68 clusters showed that the majority of these genes (78.4%, n = 1099) were effector genes, which could be classified as neuropeptides, receptors, transporters, ion channels, synaptic proteins, and cell adhesion molecules (*Figure 1—figure supplement 2G*). This result suggests the importance of effector gene profiles in brain cell classification, which has also been appreciated by previous studies in different species (*Paul et al., 2017*, *Hodge et al., 2019*).

We thus generated the hierarchical classification of all 68 cell clusters using the profiles of these 1099 effector genes. All 68 clusters were first segregated into two groups, neuronal cells (48 clusters, 37,880 cells) and non-neuronal cells (20 clusters, 7866 cells) (*Figure 1F*). Among non-neuronal cells, oligodendrocytes (Clusters 40, 42; *olig2*[+], *sox10*[+]), microglia (Cluster 55; *apoeb*[+], *mpeg1.1*[+]), endothelial cells (Clusters 48, 57; *fxyd1*[+], *rbp4*[+]), erythrocytes (Cluster 67; *hbbe1.2*[+], *hbae5*[+]), radial astrocytes (Clusters 21, 46, 65; *cx43*[+], *glua*[+]), and neuroprogenitors (Clusters 8, 14, 18, 28, 29, 33, 34, 37, 53, 56; *pcna*[+], *cdk1*[+]) were identified according to putative marker genes (*Figure 1—source data 5*). On the other hand, among neuronal cells, the first segregation defined three classes of neurons (branch I, cerebellum and habenula; branch IIa, glutamatergic neurons, branch IIb, inhibitory neurons). Branch I consisted of granule cells (Clusters 19), torus longitudinals (Cluster 45), cranial ganglions (Clusters 19, 50, 63), dorsal and ventral habenula neurons (Clusters 35 and 59). Branch IIa included 22 subclasses of excitatory glutamatergic neurons (*vgluta2a*[+]-*gad1b*[-]-*glyt2*[-]), whereas Branch IIb inhibitory neurons

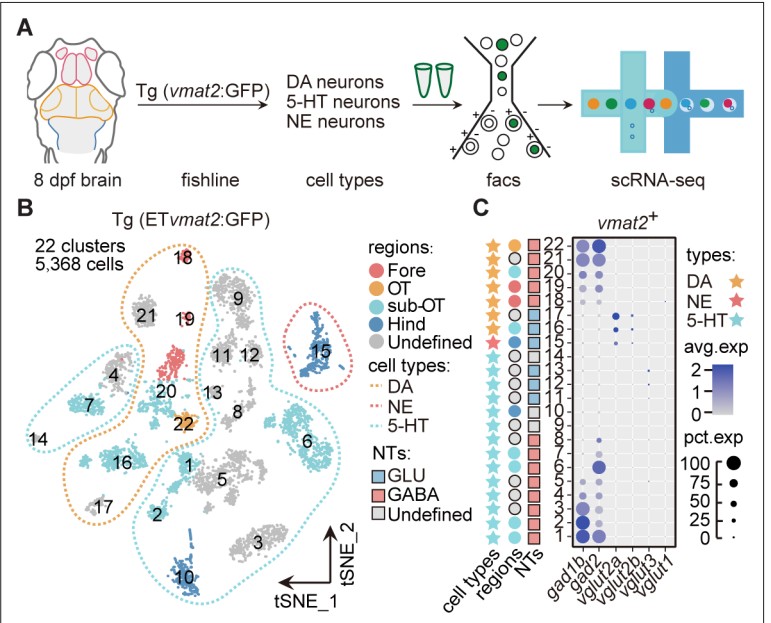

**Figure 2.** Molecular classification of neuromodulator-type neurons. (**A**) The schematic showing the procedure of collecting single-cell transcriptomes of neuromodulator neurons with fluorescence-activated cell sorting (FACS). Using Tg (ET*vmat2*:GFP) fishline, we could isolate dopaminergic (DA), serotonergic (5-HT), and norepinephrinergic (NE) neurons. (**B**) The t-distributed stochastic neighbor embedding (t-SNE) plot of 5368 cells obtained from Tg (ET*vmat2*:GFP) fishline expressing monoaminergic neuromodulators, showing 22 clusters, each marked by a number, color-coding brain regions and cell type of each cluster. (**C**) Dot plot showing the expression of glutamatergic marker (*vglut2a/vglut2b/vglu1/vglu3*) and GABAergic marker (*gad1b/gad2*) in each of the 22 *vmat2*[+] clusters in B. The neurotransmitter phenotypes were color-coded. Empty squares depict the ones with undefined neurotransmitter phenotypes. The average expression levels of these genes for all cells in each cluster were coded by the gray level. The percentage of cells expressing each gene within each cluster was coded by dot size.

The online version of this article includes the following source data and figure supplement(s) for figure 2:

**Source data 1.** The annotation of neuromodulator-type neuronal types with well-known markers.

**Figure supplement 1.** Molecular classification of neuromodulator neurons.

included 17 subclasses of GABAergic neurons ($vgluta2a^-/gad1b^+$) and 1 subclass of glycinergic neurons ($vgluta2a^-$-$gad1b^+$-$glyt2^+$, Cluster17, *Figure 1—source data 5*).

## Molecular classification of neuromodulator-type neurons

To further examine the expression of neuromodulators at the whole-brain level, we performed the scRNA-seq analysis of neuromodulator neurons sorted from the whole brain of Tg (ET*vmat2*:GFP) transgenic fish (*Figure 2A*). Using this fishline, we could examine transcriptomes of DA neurons, 5-hydroxytryptamine (5-HT) neurons, and norepinephrinergic (NE) neurons (*Wen et al., 2008*). After the filtering procedures described above, we obtained a total of 5368 *vmat*2-expressing cells (Materials and methods). The analysis aggregated these cells into 22 clusters in t-SNE plots (*Figure 2B*), more than those found using whole-brain samples (two DA, one 5-HT, and three ChAT; *Figure 1—source data 2*). To further validate the stability of clusters, we used the Jaccard index, which showed that the majority of clusters (20/22) were stable using mean/median Jaccard index >0.6 as cutoff (*Figure 2—figure supplement 1A*).

According to region-specific marker genes identified above and known marker genes for neuromodulator neurons, we assigned seven clusters (Fore: Clusters 18–19; sub-OT: Clusters 16, 20; OT: Cluster 22; and Clusters 17, 21 without specific regional identity) as DA neurons, 14 clusters (Hind: Cluster 10; sub-OT: Clusters 1, 2, 6, 7; and Clusters 3–5, 8–9, 11–14 without regional identity) as 5-HT neurons, one cluster (Hind: Cluster 15) as NE neurons (*Figure 2—figure supplement 1C-D*, *Figure 2—source data 1*).

Further examination of these neuromodulator clusters for their expression of specific neurotransmitters showed that the majority of 5-HT (8/14) and DA clusters (5/7) expressed GABAergic markers *gad1b/gad2* (*Figure 2C*). This result was further validated by Tg (ET*vmat*2:GFP::*gad1b*:gal4::*uas*:mCherry) (*Figure 2—figure supplement 1E*). Only three 5-HT clusters (Clusters 11–13) expressed glutamatergic marker *vglut3*, two DA clusters (Clusters 16, 17), and one NE cluster (Cluster 15) expressed glutamatergic markers, *vglut2a* and *vglut2b* (*Figure 2C*). These results are consistent with previous studies (*Filippi et al., 2014*). Besides, we also found three clusters (Clusters 24, 41, and 64) in whole brain showed choline and glutamate preferential co-expression (*Figure 1—source data 2*). Thus, our analysis provided a whole-brain characterization of the co-expression patterns of neurotransmitters and neuromodulators. In sum, DA and 5-HT neurons preferentially expressed GABAergic markers, whereas NE and ChAT neurons mostly expressed glutamatergic markers.

## The TF regulatory landscape of whole-brain glutamatergic/GABAergic neuron clusters

In the hierarchical classification based on effector gene profiles (*Figure 1F*), glutamatergic/GABAergic clusters (n = 39) at the terminus pairs represented the ones with the most similar terminal features, termed 'sister clusters'. In addition, the certainty of this hierarchical classification was verified by the bootstrap re-sampling analysis using pvclust v.2.0 (*Figure 3—figure supplement 1A*; *Suzuki and Shimodaira, 2006*). We identified 11 pairs of sister clusters, neurons in each pair exhibited the same neurotransmitter types (*Figure 3—figure supplement 2A*). To our surprise, neurons of each sister clusters could be from either the same (n = 6) or different (n = 5) brain regions (*Figure 3—figure supplement 2A*), which did not reflect the strong brain region preference.

To further examine the TF profiles of these effector gene-based sister clusters, we classified glutamatergic (IIa) and inhibitory (IIb) neurotransmitter-type neurons using TF profiles. Notably, TF-based and effector gene-based trees were distinct in terms of matching node (only one matching node: Clusters 9/61) and tree distances (tree distance = 0.71, *Figure 3—figure supplement 1B*), suggesting that the effector gene-based sister clusters (*Figure 3—figure supplement 2A*) might express different TF profiles.

We found out of 11 effector genes-based sister clusters, only one pairs could be found in TF-based sister clusters (Clusters 9/61, 'matched pattern', *Figure 3—figure supplement 2B*). And other 10 effector gene-based sister clusters were separated in TF-based classification, suggesting that neurons with similar terminal features mostly expressed different TF profiles ('convergent pattern'; *Figure 3—figure supplement 2C*). Also, neurons in each of these 10 sister clusters could come from either the same (n = 5) or different (n = 5) brain regions, exhibiting no brain region preference (*Figure 3—figure supplement 2C*).

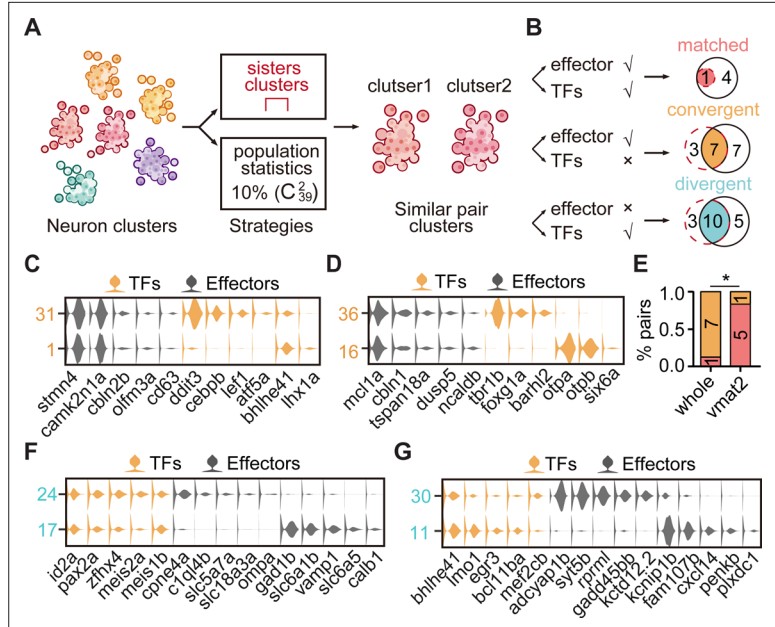

**Figure 3.** The transcription factor (TF) regulatory landscape in whole-brain neuronal clusters. (**A**) Schematic showing the strategies to assess the cluster similarity based on effector gene and TF profiles. We focused on clusters of whole-brain glutamatergic/GABAergic neurons and neuromodulator neurons. Similar pair clusters were identified by two strategies: the first strategy was based on hierarchical sister clusters. The second strategy was based on population-level statistical analysis, in which we calculated and ranked the distances of every two clusters from 39 neurotransmitter-type clusters ($C_{39}^2$) and chose the ones with the lowest 10% distance as similar pair clusters. (**B**) Left: schematic showing the criteria of three patterns: pair clusters that were similar in both TF and effector gene profiles as 'matched pattern', those pair clusters that were similar in effector gene profiles but not in TF profiles as 'convergent pattern', and those pair clusters that were similar in TF profiles but not in effector gene profiles as 'divergent pattern'. Right: the plot showing the number of each pattern using two strategies in A. The red dashed circle showing the number of cluster pairs with given pattern based on hierarchical sister cluster analysis; the black solid circle showing the number of cluster pairs with given pattern based on population-level statistical analysis. (**C–D**) Violin plots showing the expression of TFs (yellow) or effector genes (black) in glutamatergic and GABAergic similar pair clusters of convergent pattern. (**E**) The bar plot showing the proportions of different patterns for neuronal clusters with neurotransmitter or neuromodulator types. The numbers of each pattern were indicated. Fisher's exact test was used to test the significant association of different patterns, p = 0.02564, *p < 0.05. (**F–G**) Violin plots showing the expression of TF profiles (yellow) or effector gene profiles (black) in neuronal clusters of divergent pattern.

The online version of this article includes the following figure supplement(s) for figure 3:

**Figure supplement 1.** Hierarchical clustering analysis of whole-brain glutamatergic/GABAegric neurotransmitter-type neurons based on effector gene and transcription factor (TF) profiles.

**Figure supplement 2.** The transcription factor (TF) regulatory landscape in whole-brain glutamatergic/GABAergic neuronal clusters.

**Figure supplement 3.** Similar pair clusters of neuromodulator-type neurons.

**Figure supplement 4.** Divergent pattern of glutamatergic/GABAergic neurotransmitter-type neuronal clusters.

Alternatively, we performed the population-level statistical analysis to compare the landscape of TF and effector gene expression accounting for the full spectrum of cell types rather than just the most similar sister clusters. For all glutamatergic/GABAergic neuronal clusters (n = 39), we calculated the distances between every two clusters ($C_{39}^2$) based on either effector gene profiles or TF profiles, and then defined the pairs, which had the lowest 10% distances after ranking, as similar pair clusters (*Figure 3A*). Then, we defined paired clusters that were similar in both TF and effector gene profiles as 'matched pattern', those paired clusters that were similar in effector gene profiles but not in TF profiles as 'convergent pattern' (*Figure 3B*). The population-level analysis identified 19 pairs of effector gene-based similar pair clusters, 5 with matched pattern and 14 with convergent pattern

(*Figure 3—figure supplement 2D-G*). Overall, similar pair clusters with either matched or convergent pattern identified by the population-level analysis showed an overlapping but distinct pattern with those identified by the hierarchical sister cluster analysis (*Figure 3B*). This discrepancy was likely due to the following facts: (1) In the population-level statistical analysis, we arbitrarily set the lowest threshold as a criterion to identify similar pair clusters. Thus, the levels of this threshold could influence the production of similar pair clusters. (2) In population-level statistical analysis, each cluster could use for multiple times, whereas in hierarchical sister cluster analysis, once a cluster was selected as a pair with another cluster, it could not be re-used again. To overcome this discrepancy, we intersected the results from hierarchical sister cluster analysis and population-level statistical analysis, and identified eight pairs of effector gene-based glutamatergic/GABAergic similar pair clusters, one with matched pattern and seven with convergent pattern (*Figure 3B*).

We further validated these patterns of paired neuronal clusters by subsampling of genes (80% of total either TFs or effector genes) and average statistics over 20 times to re-identify similar paired clusters based on either TFs or effector genes (Materials and methods). Notably, re-identified paired clusters of different patterns completely recapitulated those pairs identified using the population-level statistical analysis above (*Figure 3—figure supplement 2L*), indicating the robustness of these patterns.

By combining sister cluster analysis and the population-level statistical analysis, we identified one paired clusters of 'matched pattern', seven paired clusters of 'convergent pattern' (*Figure 3B*). Here were some representative cases with the convergent pattern. The first case was a glutamatergic pair cluster from different brain regions: tectal glutamatergic Cluster 1 and hindbrain glutamatergic Cluster 31 shared effector gene profiles including *camk2n1a/stmn4/cbln2b/olfm3a/cd63*, but differentially expressed TF profiles, *atf5b/bhlhe41/lhx1a* and *ddit3/cebpb/lef1*, respectively (*Figure 3C*). The second case was GABAergic pair cluster from different brain regions: GABAergic clusters in the forebrain (Cluster 36) and the sub-OT (Cluster 16) shared effector gene profiles *mcl1a/cbln1/tspan18a/dusp5/ncaldb*, but differentially expressed TF profiles, *tbr1b/foxg1a/barhl2* and *otpa/otpb/six6a*, respectively (*Figure 3D*). Consistently, each of the above TF profiles has previously been characterized to participate in the specification of either glutamatergic or GABAergic neurons (*Li et al., 2007*; *Kala et al., 2009*; *Talbot et al., 2010*; *Waite et al., 2011*; *Achim et al., 2013*).

## The TF regulatory landscape of neuromodulatory neuron clusters

We further examined neuromodulator-type sister clusters based on either TF or effector gene profiles (*Figure 3—figure supplement 3A*). Using hierarchical sister cluster analysis, 5/8 of neuromodulator-type sister clusters matched with those defined by TF profiles ('matched pattern'), 3/8 sister cluster were found to be separated in TF-based classification ('convergent pattern', *Figure 3—figure supplement 3D-G*).

Then, we also performed the population-level statistical analysis using all genes or 80% genes (subsampling) to compare the landscape of TF and effector gene expression. We found eight pairs with matched pattern and one pair with convergent pattern (*Figure 3—figure supplement 3H-I*). Thus, we intersected hierarchical sister cluster analysis and population-level statistical analysis, and identified six pairs of similar neuromodulator pair clusters, five with matched pattern, and one with convergent pattern (*Figure 3—figure supplement 3J-L*).

Together with the above results of neuromodulator-type and neurotransmitter types, our analysis showed that neuromodulator pairs with similar effector gene profiles predominantly were 'matched' pattern, whereas neurotransmitter pairs with similar effector gene profiles mainly were 'convergent' pattern (*Figure 3E*). Moreover, we performed global tree measurement using the R package 'TreeDist', and found that the distance between TF-based and effector-based hierarchical tree of neuromodulator neurons (Tree distance = 0.38, *Figure 3—figure supplement 3A-B*) was lower than that of glutamatergic/GABAergic neurons (tree distance = 0.71, *Figure 3—figure supplement 1B*), suggesting distinct TF regulatory logic of effector-based phenotypes between neurotransmitter-type and neuromodulator-type at the global level. The smaller distance for neuromodulator neurons was consistent with our conclusion that neuromodulator-type clusters predominantly expressed the same TF profiles ('matched'). The 'matched' pattern suggested the generation of similar neuromodulator pairs shared specific TF programs ('stereotyped programming'), and the 'convergent' pattern suggested the generation of similar neurotransmitter pairs could be generated by different TF

programs ('flexible programming'). These different strategies may account for the fact that across species, neuromodulator types are more conserved, whereas neurotransmitter types are much diverse and variable (*La Manno et al., 2016*; *Saunders et al., 2018*; *Tiklová et al., 2019*; *Poulin et al., 2020*).

## Different neuronal clusters exhibit similar TF profiles

On the other hand, in glutamatergic/GABAergic neuronal classification, we surprisingly found that 13 pairs of sister clusters with similar TF profiles were separated in the effector gene-based classification. In other words, different from matched and convergent pairs mentioned above (*Figure 3B–D*), these 13 glutamatergic/GABAergic clusters exhibited different effector gene profiles but similar TF profiles, here terms as 'divergent' pattern (*Figure 3—figure supplement 4A*). Also, the population-level statistical analysis identified divergent pairs (n = 15, *Figure 3—figure supplement 4B*) that were largely overlapped with those identified by hierarchical sister analysis (n = 10, *Figure 3B*).

Neurons in each of these 10 divergent paired clusters could be from the same (n = 6) or different (n = 4) brain regions (*Figure 3—figure supplement 4A*). More strikingly, two pairs of neuronal clusters with divergent pattern expressed different neurotransmitters. For examples, sub-tectal glutamatergic neurons (cluster 24) and hindbrain GABAergic neurons (cluster 17) shared similar TF profiles *id2a/pax2a/zfhx4/meis1b*, but expressed different effector gene profiles, *cpne4a/c1ql4b/slc5a7a* and *gad1b/slc6a1b/slc6a5*, respectively (*Figure 3F*); forebrain glutamatergic neurons (Cluster 30) and tectal GABAergic neurons (Cluster 11) shared TF profiles, *bhlhe41/bcl11ba/mef2cb*, but had differentially expressed effector gene profiles, *adcyap1b/syt5b/rprml/gadd45bb*, and *kcnip1b/cxcl4/plxdc1*, respectively (*Figure 3G*). This result illustrated that neurons with different terminal features could also express the same TF profiles. Together, our analysis demonstrated the landscape of TFs in whole-brain-wide neuronal phenotypes in the larval zebrafish brain.

## Combinatorial TFs in marking tectal morphological subclasses

Neuronal morphology and effector gene profile are two critical criteria of neuron diversity classification (*Sugino et al., 2019*; *Peng et al., 2021*). Also, many previous studies have provided the apparent links between effector genes and neuron morphology (*Whitford et al., 2002*, *Marcette et al., 2014*; *Delandre et al., 2016*; *Noblett et al., 2019*; *Peng et al., 2021*). Thus, after we found three patterns ('matched', 'convergent', and 'divergent'), we wondered if similar patterns present between morphological subclasses and TFs.

To experimentally verify the relationship between TFs and neuronal phenotypes, we focused on morphological subtypes of tectal glutamatergic neurons. Higher-coverage transcriptomes of tectal glutamatergic neurons were further achieved by scRNA-seq of sorted tectal glutamatergic neurons in 8 dpf using Tg (*vglut2a*:loxp-DsRed-loxp-GFP) fishline (*Figure 4A*). Canonical correlation analysis (CCA) of cells from two independent experiments indicated the reproducibility of the analysis (*Figure 4—figure supplement 1A*). Analysis of 3, 883 sorted tectal glutamatergic neurons yielded 11 clusters (n = 11, *Figure 4A*), more than those identified from the whole-brain samples (n = 5, *Figure 4—figure supplement 1B*). The majority of clusters (10/11) were stable using mean/median Jaccard index >0.6 as cutoff (*Figure 4—figure supplement 1D*). Each cluster of tectal glutamatergic neurons was identified by TF marker genes (*Figure 4—figure supplement 1E-F*, *Figure 4—source data 1*).

We developed a labeling strategy using three different plasmids to analyze morphological subclasses of tectal glutamatergic neurons expressing identified TFs. First, we placed Gal4FF cassette at the start-codon of each TF by bacterial artificial chromosome (BAC) recombination technique (*Suster et al., 2011*) and constructed the Gal4FF plasmids for 15 marker TF genes (*Figure 4B*, Materials and methods). Second, we generated *vglut2a*:CRE BAC plasmid using a similar method. Third, plasmid *uas*:loxp-stop-loxp-tdTomatocaax was used with the first two plasmids to intersectionally label single tectal glutamatergic neurons, each expressing one specific TF (*Figure 4C*). These labeled neurons were then used for morphological analysis. We labeled and defined morphological subclasses of tectal neurons by injecting all three plasmids into zebrafish embryos at 4–16 cells stage. We observed the enormous variability in the morphology of tectal glutamatergic neurons in their more refined structures. By the limited number of neurons analyzed (n = 574), we were unlikely to define morphological subclasses using a global morphological description. Instead, we used the criterion based on major morphological features, including stratification, soma position, and projection patterns, to define the morphological subclasses in the current study. In addition, similar morphological classification has also

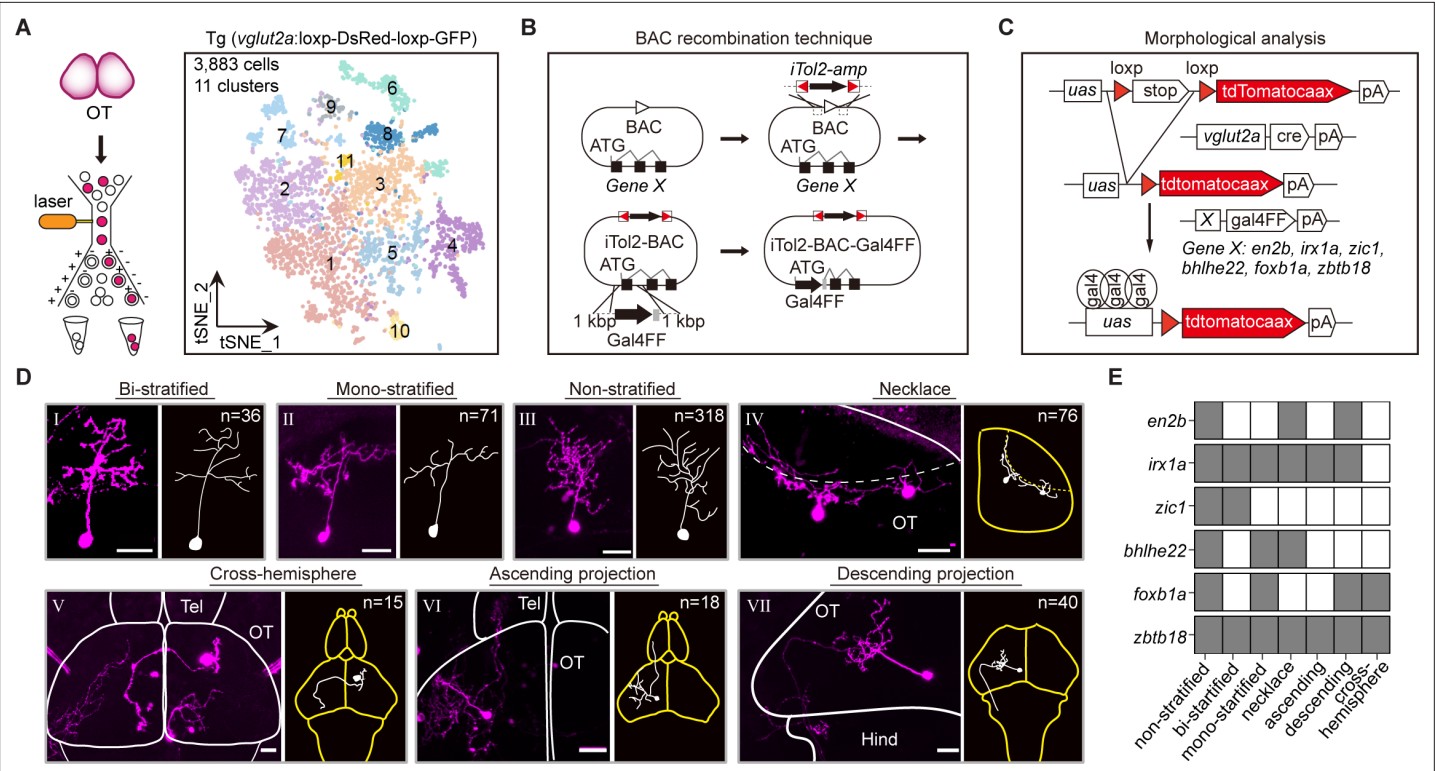

**Figure 4.** Combinatorial transcription factors (TFs) in marking tectal morphological subclasses. (**A**) Left: the schematic showing the procedure of collecting single-cell transcriptomes of tectal glutamatergic neurons with fluorescence-activated cell sorting (FACS) using Tg (*vglut2a*:loxp-DsRed-loxp-GFP) fishline. Right: the t-distributed stochastic neighbor embedding (t-SNE) plot of 3, 883 cells obtained from Tg (*vglut2a*:loxp-DsRed-loxp-GFP) fishline labelling glutamatergic neurons in the optic tectum showing 11 clusters, each color-coded and marked by a number. (**B**) Schematic showing the designing of the Gal4FF BAC plasmids for six TF marker genes, covering all 11 clusters of tectal glutamatergic neurons shown in A. (**C**) Schematic showing the method for single TF-based labeling of tectal glutamatergic neurons. CRE expressed in glutamatergic neurons was used to excise loxp and drive fluorescent tdTomatocaax expression. (**D**) Representative images of seven subclasses of tectal glutamatergic neurons with distinct morphological subclasses using the method described in (B–C). The number of neurons collected for each subclass was shown. Insets depicted the morphological characteristics. Neurons with ascending and descending projections were those projecting to the forebrain and hindbrain, respectively. Solid lines marked the boundaries between brain regions. Dashed lines marked the boundary of the tectal neuropile. Scale bars: 20 µm. (**E**) The matrix showing the expressions of six TFs in each of seven morphological subclasses. The black squares represented TFs could label particular morphological subclasses, and no expression was indicated by white squares.

The online version of this article includes the following source data and figure supplement(s) for figure 4:

**Source data 1.** The annotation of transcription factors (TFs) expression in each tectal glutamatergic clusters.

**Source data 2.** Gene labeled morphological analysis of tectal glutamatergic neurons.

**Figure supplement 1.** Combinatorial transcription factors (TFs) in marking tectal morphological subclasses.

been used previously for tectal neurons (*Nevin et al., 2010*, *Robles et al., 2011*; *DeMarco et al., 2020*). We selected six TF marker genes (*en2b*, *foxb1a*, *zic1*, *bhlhe22*, *zbtb18*, and *irx1a*) for further analysis based on two criteria: (1) These TFs are highly expressed and specific to individual tectal glutamatergic clusters based on scRNA-seq analysis. (2) Their BAC plasmids could reliably mark particular morphological subclasses (at least in four animals).

Furthermore, using confocal imaging, we reconstructed a total of 574 tectal neurons (from 263 zebrafish, *Figure 4—source data 2*) that expressed one of these six TFs (*Figure 4—figure supplement 1G*). These neurons were categorized into seven morphological subclasses: bi-stratified (I, n = 36), mono-stratified (II, n = 71), non-stratified (III, n = 318), necklace-like (IV, n = 76), cross-hemispheric (V, n = 15), ascending (VI, n = 18), and descending (VII, n = 40) (*Figure 4D*).

Whether a specific TF could serve as a marker for a particular morphological subclass was determined by the criterion that the TF appeared in a given subclass at least four times (from at least four fish). Remarkably, all six TFs marked multiple morphological subclasses in a combinatorial manner,

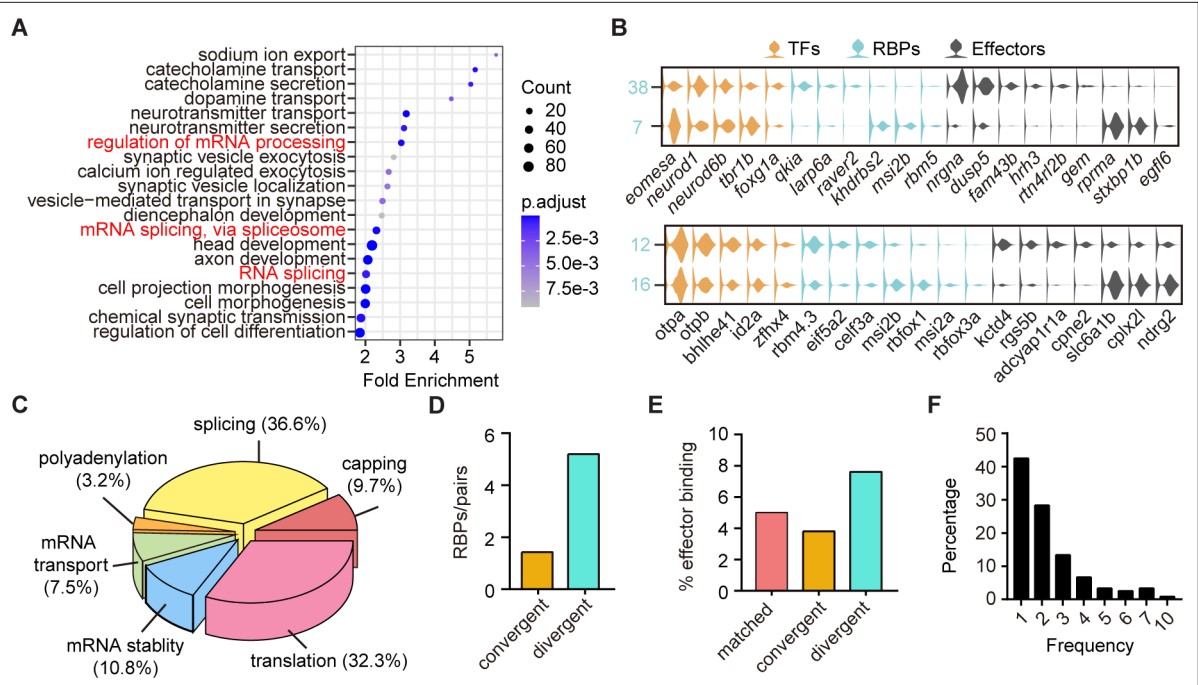

**Figure 5.** The post-transcriptional regulatory landscape in neurons with different terminal features but similar transcription factor (TF) profiles. (**A**) Dot plot showing Gene Ontology (GO) analysis of differentially expressed genes between each pair clusters with divergent pattern. Red highlighted the category of post-transcriptional regulators. The number of genes in each category was coded by dot size, p.adjust of each category was color-coded. (**B**) Violin plots showing the expression of genes encoding TFs (yellow), RNA-binding proteins (RBPs) (blue), and effector genes (black) in two pairs of neuronal clusters with divergent pattern (7/38 were glutamatergic neurons, 12/16 were GABAergic neurons). (**C**) 3D pie plot showing the functional annotation of differentially expressed RBP genes identified in A. (**D**) Bar plot showing the number of pattern-specific RBPs per pair in different patterns. Each pattern was color-coded. (**E**) Bar plot showing the effector binding target proportion of pattern-specific RBPs. Each pattern was color-coded. (**F**) Bar plot showing the percentage of one RBP used as differentially expressed genes between pair clusters with different patterns.

The online version of this article includes the following source data and figure supplement(s) for figure 5:

**Source data 1.** Differentially gene expression between sister pair clusters with matched, convergent, and divergent patterns.

**Source data 2.** The p-value of differential gene expression between sister pair clusters.

**Source data 3.** Annotation the function of RNA-binding proteins (RBPs).

**Source data 4.** The binding sites of pattern specific RNA-binding proteins (RBPs).

**Source data 5.** The combinatorial usage of RNA-binding proteins (RBPs).

**Figure supplement 1.** The post-transcriptional regulatory landscape in neurons with different terminal features but similar transcription factor (TF) profiles.

while each of seven morphological subclasses was marked by numerous TFs (*Figure 4E*). For example, *zic1* highly marked non-stratified and bi-stratified subclasses, and all six TFs marked the non-stratified subclass.

In summary, we found that single TF could mark multiple morphological subtypes in the optic tectum, and multiple TFs could mark a single morphological subtype. This observation could be inferred from 'convergent pattern' (same TFs were expressed in different effector-based subtypes) and 'divergent pattern' (different TFs were expressed in similar effector-based subtypes), respectively. However, we could not exclude unknown indirect regulations of morphological subtypes by TFs.

## The landscape of post-transcriptional regulators in neuronal clusters with different terminal features but similar TF profiles

The above finding that neuronal clusters with different terminal features expressed similar TF profiles were likely resulted from non-TF determinants (*Figure 3F–G*). We then performed GO analysis of differentially expressed genes between sister clusters of 10 pairs with different terminal features but similar TF profiles ('divergent pattern'). We surprisingly found that many differentially expressed genes

encoded RBPs (*Figure 5A*). We thus yielded a library of RBP-encoded genes that could differentially mark neurons with different terminal features (*Figure 5—source data 1*). Here were two examples: two forebrain glutamatergic neuronal clusters (Clusters 7 and 38) shared the similar expression pattern of TF profiles *eomesa/neurod1/tbr1b/foxg1a*, but had different effector gene profiles and RBPs, *rprma/stxbp1b/egfl6/khdrbs2/msi2b/rbm5* and *nrgna/susp5/fam43b/qkia/larp6a*, respectively; GABAergic neurons in hindbrain (Cluster 12) and sub-tectal (Cluster 16) shared TF profiles, *otpa/bhl-he41/id2a*, but differentially expressed effector genes and RBPs, *kctd4/rgs5b/cpne2/eif5a2/celf3a* and *slc6a1b/cplx2l /ndrg2/msi2b/rbfox1/msi2a*, respectively (*Figure 5B*). These RBP-encoded genes exhibited wide RNA functions, including capping, splicing, polyadenylation, transport, stability, and translation (*Figure 5C*, *Figure 5—source data 3*).

Besides, our analysis also yielded a comprehensive list of genes encoding RBPs marking other patterns (*Figure 5—figure supplement 1A-D*, *Figure 5—source data 1*, *Figure 5—source data 2*). We found that some RBPs exhibited differential expression between paired clusters specific to one of the above three patterns (*Figure 5—figure supplement 1*). Also, we found that the number of RBPs that were specific to divergent pattern (n = 52 from 10 pairs) was much more than those specific to either matched pattern (n = 7 from one pair) or convergent pattern (n = 10 from seven pairs, *Figure 5—figure supplement 1E-F*). Thus, in terms of the number of pattern-specific RBPs per pair, RBPs showed the preferential expressions in divergent pairs compared to convergent ones (*Figure 5D*). Note that since there was only one pair of matched pattern, we did not include it in this analysis. These pattern-specific RBPs were known to be involved in a wide range of RNA functions, including capping, splicing, polyadenylation, transport, stability, and translation (*Figure 5—figure supplement 1F*, *Figure 5—source data 3*). Thus, in addition to TFs, post-transcriptional regulatory factors also play significant roles in determining transcriptome profiles of neuronal classification.

Furthermore, we examined effector gene that were targeted by pattern-specific RBPs using the oRNAment database (http://rnabiology.ircm.qc.ca/oRNAment/) to explore a potential causality between the divergence of RBPs and the divergence of effector genes (*Benoit Bouvrette et al., 2020*). GO analysis showed that pattern-specific RBPs targeted various molecular categories including effector genes, TFs, mRNA processing, metabolism (*Figure 5—figure supplement 1H-J*, *Figure 5—source data 4*). More importantly, divergent pattern-specific RBPs targeted significantly higher proportions of effector genes than matched and convergent pattern-specific RBPs (*Figure 5E*). These results suggested the potential causality between RBPs and effector gene profiles in divergent pairs.

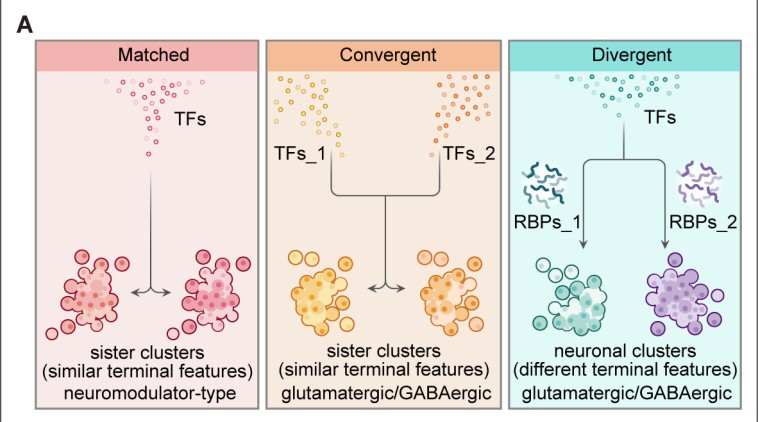

**Figure 6.** Organization of transcriptional and post-transcriptional regulators in the specification of brain-wide neuronal clusters. (**A**) Graphic summary describes the general organization of transcription factors (TFs) and post-transcriptional regulators in the specification of neuronal clusters at the whole-brain level. Effector genes describe the terminal features of whole-brain neuronal classification. Neuronal clusters with similar terminal features (named 'sister clusters') can share TF profiles ('matched', left), or exhibit distinct TF profiles ('convergent', middle). Besides, neuronal clusters with distinct effector gene profiles (terminal features) can share TFs profiles ('divergent', right), and in each pairs, two neuronal clusters are differentially marked by the expression of RNA-binding proteins (RBPs) at the post-transcriptional levels. Moreover, sister clusters with 'convergent' and 'divergent' patterns are glutamatergic/GABAergic neurons, whereas sister clusters with 'matched' pattern are neuromodulator-type neurons.

Interestingly, genes encoding well-known sequence-specific RBP Rbfox1-encoded gene was differentially expressed in multiple neuronal clusters, like 23/25, 27/54, 11/30, 13/24, 12/16, 2/3, and 7/38 (*Figure 5—source data 5*). Rbfox1 specifically recognizes UGCAUG motifs, which are often found at 5′- and 3′-regions of introns (*Yui Jin, 2003*; *Vuong et al., 2018*). Previous studies have elegantly revealed the importance of activity-dependent splicing regulator Rbfox1 in the transition from neuroprogenitors to neurons as well as in the interneuron subtype-specific splicing in the mouse cortex (*Zhang et al., 2016*). Moreover, we calculated the frequency of individual RBPs in marking all sister pair clusters with each of three patterns and found that single RBPs were frequently present in multiple pair clusters (42.5%, 51/120, *Figure 5F*), indicating that clusters mostly used RBPs in a combinatorial manner.

Thus, our analysis revealed the importance of RBP-encoded genes in specifying diverse neuronal phenotypes, and the identification of a comprehensive list of RBP-encoded genes in our study provides a valuable resource for future investigation.

## Discussion

This study analyzed ~65,000 single cells from the whole-brain, specific brain regions, neurotransmitter/neuromodulator-defined neuronal populations of the 8 dpf larval zebrafish (*Figures 1A, 2B , and 4A*). Notably, our transcriptome analysis offered a close-to-full coverage of all cells (~100,000) in the zebrafish brain, providing multidimensional landscapes of TFs and post-transcriptional regulators in vertebrate whole-brain neuron classification (*Figure 6*).

A significant focus of our analysis is the analysis of sister clusters at the termini of effector gene-based hierarchical classification (*Figure 3A*). These sister pair clusters represent the finest molecularly defined neuronal subclasses based on effector gene profiles. Because sister clusters of the terminus pair with highly similar terminal features, it offers us an opportunity to study the determinants of their terminal feature similarity and factors that lead to their slight diversification. Our analysis of the expression profiles of regulatory genes in sister clusters including TFs and post-transcriptional regulators (*Figure 6*) may thus help to elucidate the molecular logic underlying neuronal diversification.

Specifically, our analysis identified three patterns of TFs in specifying neuronal phenotypes, 'matched pattern', 'convergent pattern', and 'divergent pattern' (*Figures 3B and 6A*). Glutamatergic/GABAergic sister clusters with similar terminal features mostly expressed distinct TF profiles ('convergent pattern'), whereas neuromodulator-type sister clusters largely expressed the same TF profiles ('matched pattern', *Figure 3E*). The results indicated that the 'matched' pattern supports the notion that the same set of TFs plays a crucial role in determining a given profile of neuromodulator-type terminal features, whereas the 'convergent' pattern suggests that even though TFs are important for subclass determination, different TFs could still converge onto the same transcriptome phenotype. Combined with earlier findings of extensive phenotypic convergence of distinct neuron types in worms and flies (*Gendrel et al., 2016*; *Konstantinides et al., 2018*; *Hobert and Kratsios, 2019*), our findings in zebrafish strongly suggest that phenotypic convergence is highly conserved from invertebrates to vertebrates.

More interestingly, we identified 10 terminus clusters exhibiting different transcriptome patterns but similar TF profiles ('divergent pattern'). Thus, these clusters are likely determined by factors other than TFs (*Figure 3F–G*). Our further analyses revealed the potential importance of the expression pattern of post-transcriptional regulators, particularly RBPs, in marking these neuron clusters (*Figure 5A*). For instance, *upf3a* (the marker distinguishing subclasses of the pairs 23/25, 12/16, and 60/62, *Figure 5—source data 5*) is known to regulate mRNA stability via nonsense-mediated RNA decay (*Shum et al., 2016*), *rbfox1/rbfox3a* (the marker distinguishing subclasses of the pair 27/54, 13/24, 11/30, 23/25, 12/16, 2/3, and 7/38, *Figure 5—source data 5*) could mediate cell-type-specific splicing in cortical interneurons, assembly of axon initial segment, and synaptic transmission (*Jacko et al., 2018*; *Vuong et al., 2018*; *Wamsley et al., 2018*). In addition to post-transcriptional regulators, other factors could also be expected to involve neuron diversification, such as epigenetic regulators, translational regulators, protein stability, which are interesting to explore in future studies. Together, our findings suggest that combinatorial TFs and post-transcriptional regulators could work in concert to determine neuronal types in the larval zebrafish brain (*Figure 6*).

Regional identity is an essential factor for classifying brain cells. In the larval zebrafish brain, our results showed that neurons, qRG, and neuronal progenitors exhibited prominent regional

characteristics (*Figure 1C*). In the hierarchical clustering and population-level statistical analysis, similar pair clusters from the same region or different regions occur in nearly equal probabilities (*Figure 3—figure supplement 2A*). Considering the fact that neuronal phenotypes from the same region could exhibit common regional identities, it was surprising to observe that such a higher proportion of sister subclasses with similar transcriptomic phenotypes could arise from two different brain regions. Putting this finding into the context of neuron-type evolution, it raises the possibility that different brain regions independently give rise to similar neurotransmitter phenotypes through different TF programs. Alternatively, as brain regions are functionally diversified during the evolution, these highly similar neuronal clusters of different brain regions possibly derive from ancient building blocks, which become divergent through the evolutionary acquisition of different combinatorial TF codes.

scRNA-seq resolves the transcriptomes of brain cells at a given time point (*Erhard et al., 2019*). On the other hand, gene expression is highly dynamic and sensitive to spontaneous or stimulus-evoked neuronal activities (*West et al., 2002*; *Kim et al., 2010*; *Yap and Greenberg, 2018*), circadian rhythm (*Panda et al., 2002*, *Takahashi, 2017*), as well as other extrinsic and intrinsic factors. Thus, gene profiles of individual neuron subclasses could represent dynamic cellular states at a given time rather than a static subclass (*Wu et al., 2016*, *Ofengeim et al., 2017*). This issue could be resolved by comparing transcriptome profiles of cell samples from different animals or from different developmental stages. In this study, the 68 cell subclasses that we have identified for the larval zebrafish (8 dpf) were confirmed in another fish brain of the same age and were largely recapitulated by cell subclasses identified from the juvenile zebrafish (23–25 dpf, *Figure 1E* and *Figure 1—figure supplement 2F*; *Raj et al., 2018*). Thus, transcriptome-defined subclasses in our study indeed largely represent stable neuronal subclasses. An alternative demonstration of stable neuronal subclasses is to perform genetic labeling of the brain using promoters of marker genes for transcriptome-defined subclasses. Reproducible labeling of the same morphological subclasses, as demonstrated in our morphological studies of tectal neurons expressing specific TFs, also supports that we have identified stable neuronal subclasses (*Figure 4D–E*). Identification of neuronal subclasses in the present study paves the way for understanding neuronal diversity. Future studies of temporal changes of single-cell transcriptome profiles in the whole brain could provide important insights into the dynamic changes of the brain.

# Materials and methods

**Key resources table**

| Reagent type (species) or resource | Designation | Source or reference | Identifiers | Additional information |
|---|---|---|---|---|
| Strain, strain background (*Danio rerio*) | Wild type | Dr William | | AB |
| Strain, strain background (*Danio rerio*) | gad1b:EGFP | *Wang et al., 2020* | ZDB-TGCONSTRCT-210507–9 | Tg (gad1b:EGFP) |
| Strain, strain background (*Danio rerio*) | vglut2a:loxp-DsRed:loxp-GFP | *Satou et al., 2012* | ZDB-FISH-150901–9050 | Tg (vglut2a:loxp-DsRed:loxp-GFP) |
| Strain, strain background (*Danio rerio*) | glyT2:GFP | *McLean et al., 2007* | ZDB-ALT-070514–1 | Tg (glyT2:GFP) |
| Strain, strain background (*Danio rerio*) | vmat2:GFP | *Wen et al., 2008* | ZDB-PUB-080102–11 | Tg (ETvmat2:GFP) |
| Strain, strain background (*Danio rerio*) | elavl3: H2B-GCaMP6s | *Freeman et al., 2014* | ZDB-TGCONSTRCT-190827–1 | Tg (elavl3: H2B-GCaMP6s) |
| Recombinant DNA reagent | pTol2-10xUAS:loxp-stop-loxp-tdtomato | This paper | | We made this plasmid by ligated three PCR fragments: loxp-stop-loxp, tdTomatocaax and 10× uas backbone |

*Continued on next page*

*Continued*

| Reagent type (species) or resource | Designation | Source or reference | Identifiers | Additional information |
|---|---|---|---|---|
| Recombinant DNA reagent | vglut2a:cre | This paper | | BAC plasmid use BAC (CH211-111D5) |
| Recombinant DNA reagent | zic1:gal4FF | This paper | | BAC plasmid use BAC (CH211-95F4) |
| Recombinant DNA reagent | bhlhe22:gal4FF | This paper | | BAC plasmid use BAC (CH211-277b21) |
| Recombinant DNA reagent | en2b:gal4FF | This paper | | BAC plasmid use BAC (DKEY-265A7) |
| Recombinant DNA reagent | foxb1a:gal4FF | This paper | | BAC plasmid use BAC (CH211-2C17R) |
| Recombinant DNA reagent | zbtb18:gal4FF | This paper | | BAC plasmid use BAC (CH211-221N23) |
| Recombinant DNA reagent | irx1a:gal4FF | This paper | | BAC plasmid use BAC (CH73-211K12) |
| Commercial assay or kit | Single Cell 3' Library and Gel Bead kit v2 Chip kit | 10× Genomics | 120237 | scRNA-seq |
| Commercial assay or kit | dsDNA High Sensitivity Assay Kit | AATI | DNF-474–0500 | scRNA-seq |
| Commercial assay or kit | ClonExpressMultiS One Step Cloning Kit | Vazyme | Cat#C112-01/02 | Prepare recombinant plasmid |
| Chemical compound, drug | papain | Worthington Biochemical Corporation | LS003126 | Prepare papain solution to dissociation cells |
| Chemical compound, drug | DNase I | Sigma | Cat#DN25 | Prepare papain solution to dissociate cells |
| Chemical compound, drug | L-cysteine | Sigma | Cat#C6852 | Prepare papain solution to dissociate cells |
| Chemical compound, drug | DMEM/F12 | Invitrogen | Cat#11330032 | Prepare papain solution to dissociate cells |
| Chemical compound, drug | 45% glucose | Gibco | Cat#04196545SB | Prepare wash buffer during dissociation |
| Chemical compound, drug | HEPES | Sigma | Cat#H4034 | Prepare wash buffer during dissociation |
| Chemical compound, drug | FBS | Gibco | Cat#10270106 | Prepare wash buffer during dissociation |
| Chemical compound, drug | DPBS | Invitrogen | Cat#14190–144 | Prepare wash buffer during dissociation |
| Chemical compound, drug | MS222 | Sigma | Cat#A5040 | Anaesthesia |
| Chemical compound, drug | Low melting agarose | Sigma | Cat#A0701 | Embedded fish |
| Software, algorithm | R 3.5.1 | R-project | https://www.r-project.org/ | Data analysis |
| Software, algorithm | Cell Ranger Single Cell Software Suite (v2.1.0) | 10× Genomics | https://support.10xgenomics.com | scRNA-seq data analysis |
| Software, algorithm | Seurat | https://satijalab.org/seurat/ | http://satijalab.org/seurat/ | scRNA-seq data analysis |
| Software, algorithm | bioDist | https://www.bioconductor.org | https://www.bioconductor.org/packages/release/bioc/html/bioDist.html | |

*Continued on next page*

*Continued*

| Reagent type (species) or resource | Designation | Source or reference | Identifiers | Additional information |
|---|---|---|---|---|
| Software, algorithm | scclusteval | R-package, Jaccard index | https://github.com/crazyhottommy/scclusteval; *Tang et al., 2020* | Jaccard index could be used to evaluate the robustness of clusters |
| Software, algorithm | TreeDist | R-package, calculate two tree similarity | https://github.com/ms609/TreeDist; *Smith, 2021* | Calculate tree distance |
| Software, algorithm | FIJI | PMID:22743772 | http://fiji.sc/ | Analysis image |
| Software, algorithm | GraphPad Prism | GraphPad Software | https://www.graphpad.com | Data analysis |
| Software, algorithm | FV10-ASW 4.0 Viewer | Olympus | https://www.olympus-global.com | Analysis image |
| Software, algorithm | clusterProfiler | R-package | https://bioconductor.org/packages/release/bioc/html/clusterProfiler.html | GO analysis |
| Software, algorithm | NNLS | R-package, non-negative least squares solver from Lawson and Hanson | https://github.com/rdeits/NNLS.jl; *Deits, 2021* | Compare correlation of clusters |

## Animal maintenance and transgenic lines

For all experiments in this study, zebrafish were maintained, mated, and raised at 28°C according to standard protocols. Animals were staged according to dpf. Animal procedures performed in this study were approved by the Animal Use Committee of Institute of Neuroscience, Chinese Academy of Sciences (NA-045–2019).

Transgenic lines used in this study include: Tg (*glyT2*:GFP) (*McLean et al., 2007*), Tg (*vglut2a*: loxp-DsRed-loxp-GFP) (*Satou et al., 2012*), Tg (Et*vmat*2:GFP) (*Wen et al., 2008*), Tg (*gad1b*:EGFP) (*Wang et al., 2020*).

## Single-cell sample preparation

### Whole-brain and brain region identified cell populations

Wild-type fish were processed for 10× Genomics single-cell transcriptome sequencing. Whole brains or different brain regions (Fore, OT, Hind, and sub-OT) were dissected as anatomical structures (*Figure 1B*, *Figure 1—figure supplement 1A*). Tissues were dissociated with 300 µL papain (28 units/mL, Worthington) in papain solution (1% DNase, 12 mg/mL L-cysteine in DMEM/F12), incubated at 37°C for 15 min with proper mix methods (*Yu and He, 2019*). Dissociated cells were washed twice with washing buffer (100 mL: 650 µL 45% glucose, 500 µL 1 M HEPES and 5 mL FBS into 93.85 mL 1× DPBS, sterilized with a 0.22 µm pore size filter), and sterilized with 40 µm cell strainers (BD Falcon) into 300–400 µL washing buffer.

### Whole-brain neuromodulator or tectal glutamatergic neurons

For the transgenic fishlines, brains were dissected and dissociated as above; the cell suspension was used to sort cells with high fluorescence intensity with flow cytometry (Moflo XDP, Beckman Coulter). The interesting cells were transferred to washing buffer. In this paper, we used Tg (ET*vmat*2:GFP) transgenic fishlines to acquire neuromodulator-defined cell populations; Tg (*vglut2a*: loxp-DsRed-loxp-GFP) was used to collect glutamatergic neurons. Tg (ET*vmat*2:GFP) could label most monoaminergic neurons, including DA neurons, 5-HT neurons, and NE neurons (*Figure 2A*, *Figure 2—figure supplement 1B*; *Wen et al., 2008*).

### scRNA-seq on 10× Genomics Chromium Platform

All sampling was carried out with 10× Genomics Chromium Single Cell Kit (Version 2): suspensions prepared as described above were diluted to concentrations 300–1000 cells/mL with washing buffer, then added 10× Chromium RT mix to achieve loading target cell numbers 13,000. Downstream cDNA

synthesis (14 PCR cycles), library preparation, and sequencing were carried out according to manufacturer's instructions (https://www.10xgenomics.com/solutions/single-cell/).

## Data analysis of scRNA-seq

### Bioinformatics processing of raw reads

Sequencing data (FASTQ files) were converted to matrices of expression counts using the Cell Ranger software (Version 2.0.1) provided by 10× Genomics. Transcriptome libraries were mapped to a zebrafish reference built from a custom GTF file and the zebrafish GRCz11 (Ensemble release-96) genome assembly. For each sample, cells with less than 600 detected molecules (UMIs), or less than 1.2-fold molecule to gene ratio, and genes detected in fewer than 20 cells or more than 60% of all cells were excluded for the following analysis.

### Cell clustering analysis of cells from the whole brain

After filtering, the combined gene expression matrix of whole brain and four major brain region samples were combined (using Cellranger aggr pipeline) and loaded into Seurat package (Version 2.3.4) in R (Version 3.4.3) for the following analysis as described in tutorials (http://satijalab.org/seurat/). In brief, digital gene expression matrices were column-normalized and log-transformed. Cells with fewer than 500 expressed genes, greater than 5% mitochondrial content, very high numbers of UMIs and gene counts that were outliers of a normal distribution (likely doublets/multiplets) were removed from further analysis. Combined all samples together, we got 45,746 cells, which included four whole-brain samples, four specific brain regions (n = 2 each) including the Fore, OT, Hind, and the sub-OT; 1402 variable genes were selected for principal component analysis (PCA) by binning the average expression of all genes into 300 evenly sized groups, and calculating the median dispersion in each bin (parameters for MeanVarPlot function: x.low.cutoff = 0.0125, x.high.cutoff = 8, y.cutoff = 0.5). The top 100 principal components (PCs) were used for the first round of clustering with the Louvain modularity algorithm (FindClusters function, resolution = 2.5), which generated 68 distinct clusters (*Figure 1A*). Marker genes for each cluster were calculated using FindAllMarkers function in Seurat (parameters: min.pct = 0.1, min.diff.pct = 0.25).

### Clustering analysis of neuromodulator-defined cell populations

After filtering as above and selected all cells with neuromodulators marker gene expression (*vmat*2+ monoaminergic neurons: *th*+/*tph2*+/*th2*+/*dbh*+, *Figure 2—figure supplement 1C*, and *Figure 2—source data 1*), variable genes (n = 2,120) were selected for PCA. The top 46 PCs were used for the first round of clustering with the Louvain modularity algorithm (FindClusters function, resolution = 0.6). In total, 5398 cells obtained from Tg (ET*vmat*2:GFP) fishline expressed monoaminergic neuromodulators, showing 22 clusters (*Figure 2B*). Marker genes for each cluster were calculated using FindAllMarkers function in Seurat (parameters: min.pct = 0.1, min.diff.pct = 0.25).

### Clustering analysis of glutamatergic neurotransmitter-type cell populations

After filtering as above and selected all cells with tectal glutamatergic marker gene expression (*vglut2a*, *mab21l2*, *tubb5*, and *tfap2a*, *Figure 4—figure supplement 1C*), we obtained 3883 cells from optic tectum with Tg (*vglut2a*:loxp-DsRed-loxp-GFP) fishline expression tectal glutamatergic neurons, showing 11 clusters using 1276 variable genes and 37 PCs (*Figure 4A*). Marker genes for each cluster were calculated using FindAllMarkers function in Seurat (parameters: min.pct = 0.1, min.diff.pct = 0.25).

### Jaccard similarity index

To evaluate the stability of each cluster, we performed R package 'scclusteval', in which we re-sampled a subset of the cells (80%) from the population and repeated clustering, and then used the Jaccard index to evaluate cluster similarity before and after re-clustering. If a cluster is robust and stable, random subsampling and re-clustering will keep the cell identities within the same cluster. After repeating the re-clustering 20 times, we used the mean/median of the Jaccard index as the metric to evaluate the stability of the clusters (*Tang et al., 2020*). The distribution of the Jaccard index across subsamples measures the robustness of the cluster. Clusters with a mean/median stability score

less than 0.6 should be considered unstable (*Zumel, 2014*). Here, we used JaccardRainCloudPlot to visualize the Jaccard index and found most clusters of whole brain (*Figure 1—figure supplement 1E*), *vmat2*⁺ neuromodulator (*Figure 2—figure supplement 1A*), and tectal glutamatergic (*Figure 4—figure supplement 1D*) were stable.

### Identification of brain region markers

To identify potential brain region-specific markers that exist in all cell types, clusters were assigned when most of the cells in a cluster intermingle with cells from one particular brain region (*Figure 1—figure supplement 2B*). Then, we defined differentially expressed genes shared by all cell types (*vglut*⁺, glutamatergic neurons; *gad*⁺, GABAergic neurons; *pcna*⁺, neuroprogenitors; *cx43*⁺, radial astrocytes) to represent the targeted brain region-specific markers. We separated each cell type in four different brain regions as individual samples, FindAllMarkers function in Seurat was used to identify the differentially expressed markers of a given cell type. We listed these brain-region specific markers in *Figure 1C*, and these genes exhibited specific expression pattern in all cell types of particular brain region (*Figure 1D*). Furthermore, these brain region-specific genes were used to identify the regional origins of each cell clusters. Only clusters with over 5% of cells expressing the marker genes or with averaged UMI of genes-expressing cells over 2 were assigned regional identities (*Figure 1—source data 3*).

### The regional distribution of whole-brain neurotransmitter-type neurons

According to the marker genes that are specific to primary neurotransmitter phenotypes, including *slc17a6b* (glutamatergic neurons), *gad1b* (GABAergic neurons), and *slc6a5* (glycinergic neurons), we identified subsets of neurons with different neurotransmitter type as clusters with over 5% of cells expressing the marker genes or with averaged UMI of genes-expressing cells over 2 (*Figure 1—source data 3*). We found that the forebrain was predominantly populated by glutamatergic neurons, whereas glycinergic neurons mainly resided in the hindbrain (*Figure 1—figure supplement 2C-D*). Live image with fishline Tg (*glyT2*:GFP), Tg (*vglut2a*: loxp-DsRed-loxp-GFP), and Tg (*gad1b*:EGFP) support the results above (*Figure 1—figure supplement 2E*).

### Annotation of each cluster in neuromodulator-type neurons

To assign brain region identity for each neuromodulator-type neurons, we used markers *dlx5a*, *otpa*, *dlx2a*, *barhl2* for sub-OT, *en2a* for OT, *mab21l2* for midbrain and hindbrain, and *phox2bb*, *hoxb3a*, *mafba*, and *efnb2a* for hindbrain (*Figure 2—figure supplement 1D*). As for the subtype of neuromodulator-type neurons, we used markers *th* for DA neurons, *tph2* for 5-HT neurons, *dbh* for NE neurons, and *slc5a7a*, *chata*, *slc18a3a* as markers of ChAT neurons (*Figure 2—figure supplement 1C*). And defined clusters with over 5% of cells expressing the marker genes or with averaged UMI of genes-expressing cells over 2 as a threshold for neurons with their identities (*Figure 2—source data 1*). In addition, we used markers *gad1b*, *gad2* for GABAergic and *vglut1*, *vglut2a*, *vglut2b*, *vglut3* for glutamatergic (*Figure 2C*) as a method to assign co-expression pattern of neurotransmitter for each neuromodulator-type neurons.

### Comparison of the single-cell transcriptome between larval and juvenile zebrafish

The NNLS was applied to compare our data (8 dpf) with juvenile zebrafish (23–25 dpf, GSE105010). Top 20 (listed in *Figure 1—source data 4*) marker genes defined in our data that are also highly variable genes in juvenile data were used for decomposition. Decomposition was performed on mean expression values by averaging our dataset or by averaging other single-cell datasets using cluster assignments provided by the authors. NNLS was implemented using the 'nnls' package in R (Version 3.4.3). The 68 clusters we identified exhibited a high overlap with their counterparts in the juvenile zebrafish brain. Meanwhile, we also found clusters with specific regional origins and cell types exhibited high correlation with their juvenile counterparts (*Figure 1E* and *Figure 1—figure supplement 2F*).

## Hierarchical clustering analysis based on effector gene and TF profiles

The effector genes (n = 1099) in highly variable genes (n = 1402) were selected for the clustering analysis of whole-brain clusters, and mean expression value of each gene was calculated for each cluster. The distance matrix was defined as (1 − Pearson correlation coefficient between clusters)/2. Hclust function (clustering method: ward.D) implemented in R (Version 3.4.3) was applied for the hierarchically clustering. Six major brain cell types were identified using height = 0.42 as cutoff (shown with different color in *Figure 1A and F*). For each cell type, pan-enriched genes were identified using FindAllMarkers function in Seurat (parameter: min.pct = 0.1, min.diff.pct = 0.25). We identified these six major cell type as: branch Ⅰ (cerebellum cells and habenula cells), branch Ⅱa (glutamatergic neurons), branch Ⅱb (inhibitory neurons), branch Ⅲ (neuroprogenitors), branch Ⅳ (radial astrocytes), and branch Ⅴ (others: oligodendrocytes, microglia, and endothelia cells) according to marker genes of each branch (*Figure 1—source data 5*, *Figure 1F*).

Besides, we also selected 283 TFs in variable genes (n = 1402) to do hierarchical clustering analysis as above (*Figure 3—figure supplements 1B and 2A*). Meanwhile, we performed hierarchical analysis for *vmat2*⁺ neuromodulator neurons based on effectors (n = 1783) and TFs (n = 319) in variable genes (*Figure 3—figure supplement 3A-B*).

Moreover, R package 'pvclust' were used to assess the uncertainty in hierarchical clustering by performing bootstrap analysis of clustering (*Figure 3—figure supplement 1A*). Groups strongly supported with AU (approximately unbiased) p-value > 0.9 were highlighted.

## Identification the similarity of pair clusters using two strategies

### Hierarchical sister clusters analysis

Before analysis, we compare the similarity of tree based on effector gene and TF profiles with R package 'TreeDist'. In glutamatergic/GABAergic neurons, these two tree shown different organization, with only one matching node (node 13, Clusters 9 and 61, tree distance = 0.7127991, *Figure 3—figure supplement 1B*); while in neuromodulator neurons, these two tree shown much similar organization, with eight matching nodes (tree distance = 0.3766735, *Figure 3—figure supplement 3A-B*).

Sister clusters in the terminus of tree indicate pair cluster with most similar gene expression, we focus on 39 glutamatergic (IIa) and inhibitory (IIb) neurons and identified 11 pairs of sister clusters that exhibited high similarity in effector gene profiles and 14 sister clusters with similar TF expression (*Figure 3—figure supplement 2A*). And for *vmat2*⁺ neuromodulator neurons, we identified eight sister clusters in the terminus branches of both effector-based hierarchy tree and TF-based tree (*Figure 3—figure supplement 3C*).

### Population-level statistical analysis

To compare the landscape of TF and effector gene expression accounting for the full spectrum of cell types rather than just the most similar sister clusters, we also performed population-level statistical analysis. First, we calculate any distance of two clusters that were randomly picked from 39 glutamatergic/GABAergic neuronal clusters ($C_{39}^2$). If the distance is within the lowest 10% population of all distances containing either of two randomly selected clusters, we defined it a similar pair cluster; while the distance is within the top 80% population of all distances containing either of two randomly selected clusters, we defined it as a dissimilar pair (*Figure 3A*). Then we performed these analyses on both effector gene profiles and TFs, and defined those pair clusters similar in both effector-based and TF-based tree as 'matched pattern'; those pair clusters similar in effector-gene profiles but not TFs as 'convergent pattern', and those pair clusters similar in TF profiles but not effector-gene profiles as 'divergent pattern' (*Figure 3B*). In matched pattern, the distance of pair cluster was the lowest distance in both effector gene-based (white) and TFs-based (gray) distances (*Figure 3—figure supplement 2D*). In convergent pattern, the distance of pair cluster was the lowest distance in effector gene-based (white) distances, but not the lowest distance in TFs-based (gray) distances (*Figure 3—figure supplement 2F*). In divergent pattern, the distance of pair cluster was the lowest distance in TFs-based (gray) distance, but not the lowest distance in effector gene-based (white) distances (*Figure 3—figure supplement 4B*).

Moreover, we performed population-level statistical analysis with *vmat2*⁺ neuromodulator neurons, and found nine pairs with matched pattern which showing the lowest distance in both TF and effector

gene-based distances; two pairs with convergent pattern which showing the lowest distance in effector gene-based distance, but not in TF-based distance (*Figure 3—figure supplement 3H*).

However, this analysis is subject to the intrinsic statistics of TF and effector gene expression, as well as the sensitivity of scRNA-seq method. Thus, we intersected the results from hierarchical sister cluster and population-level statistical analysis to identify pairs with each patterns (*Figure 3B* and *Figure 3—figure supplement 3J*).

## Subsampling of genes in population-level statistical analysis

To test the robustness of three patterns based on distance measures, we subsampling 80% gene inputs in population-level statistical analysis. First, we random subsampled 80% effector genes/TFs, calculated their distance defined as (1 − Pearson correlation coefficient between clusters)/2. Then average statistics over 20 times subsampling calculations and performed population-level statistical analysis as above. These re-identified pairs completely recapitulated those identified using the population-level statistical analysis based on total genes, suggesting the robustness of three patterns in glutamatergic/GABAergic neurons (*Figure 3—figure supplement 2H-K*). All pairs re-identified are consistent with pairs with all genes (*Figure 3—figure supplement 2L*). Besides, 8 out of 12 re-identified neuromodulator-type pairs with matched pattern, 1 out of 3 re-identified pairs with convergent pattern were consistent with those identified using the population-level statistical analysis based on total genes (*Figure 3—figure supplement 3I*).

Moreover, we quantified the distance of two similar trees that was exemplified by TF-based or effector gene-based clusters with or without subsampling of 80% genes, and the result showed that the tree distances of TF-based or effector gene-based clusters before and after subsampling were only 0.20 (TF-based) and 0.14 (effector gene-based), respectively (*Figure 3—figure supplement 2H-K*). These distances are smaller than the distance between TF and effector-gene profiles (0.7127991, *Figure 3—figure supplement 1B*). All above results indicated the overall distinction of the landscape of TF-based and effector gene-based clusters.

## Comparison of two rounds of tectal glutamatergic transcriptome data

CCA in Seurat was used to test the reproducibility of two rounds of tectal glutamatergic transcriptome data. Two rounds of transcriptome data were intermingled with each other in tSNE plot showing a better performance of two parallel transcriptome data (*Figure 4—figure supplement 1A*).

## Sorted tectal glutamatergic transcriptome data yielded more clusters

The NNLS was applied to compare the transcriptome data from sorted tectal glutamatergic neurons and the subset of these neurons from whole brain. The top 20 marker genes defined in sorted sample which shared in whole-brain sample were used for decomposition. Decomposition was performed on mean expression values by averaging our dataset. NNLS was implemented using the 'nnls' package in R (Version 3.4.3). Sorted samples yield more clusters than whole-brain samples. Besides, sorted sample clusters exhibited diversity with their counterparts in the whole-brain sample (*Figure 4—figure supplement 1B*).

## Differentiation expression of genes encoding RBP between paired clusters

For each paired clusters, differential expression genes were identified using FindMarkers function in Seurat (Parameter: min.pct = 0.1, min.diff.pct = 0.25). Then we mapped the expression of genes encoding RBP within each paired clusters in different patterns (matched, convergent, and divergent patterns, *Figure 5—source data 1*, *Figure 5—figure supplement 1B-D*). Comparing the RBPs used in each pattern, we identified pattern-specific RBPs (*Figure 1—figure supplement 1E-F*).

## Searching the binding sites of pattern-specific RBPs

We take advantage of the oRNAment database (*Benoit Bouvrette et al., 2020*) to search the putative target sites of RBPs. We only found three matched pattern-specific RBPs (*celf1*, *nova1*, *pum1*), three convergent pattern-specific RBPs (*msi2a*, *taf15*, *tia1*), 18 divergent pattern-specific RBPs (*eif4a2*, *elavl1*, *hnrnpa1b*, *igf2bp3*, *hnrnpl2*, *mbnl2*, *mbnl3*, *pcbp4*, *raly*, *rbm25b*, *rbm28*, *sart3*, *sf3b4*, *sfpq*, *snrnp70*, *snrpa1*, *taf13*, *u2af2b*) in the database. Then, we searched the putative binding sites of these

RBPs use score > 0.5 as a cutoff (matched: 2987 binding sites; convergent: 2755 binding sites; divergent: 2856 binding sites, *Figure 5—source data 4*).

## Gene ontology

R package 'clusterProfiler' was used to analyze and visualize function profile of gene and gene clusters (*Yu et al., 2012*). Bioconductor annotation packages org.Dr.eg.db were imported for genome-wide annotation of mapping Entrez gene identifiers or ORF identifiers for zebrafish. We performed GO analysis on 1402 variable genes of whole-brain and 2120 variable genes of *vmat2*⁺ neuromodulator-type neurons, and found most genes belong to effector genes, including receptors, neuropeptide, ion channel, and so on (*Figure 1—figure supplement 2G*).

We performed GO analysis on differential expression genes between sister pair clusters with different patterns (convergent and divergent), and found these genes belong to TFs, effectors, and genes encoding RBP (*Figure 5A*, *Figure 5—figure supplement 1A*).

Besides, we performed GO analysis of binding target sites of pattern-specific RBPs, showing the TFs, effectors, mRNA processing are the major targets in these RBPs (*Figure 5—figure supplement 1H-J*). And found the percentage of effectors in all binding sites of divergent pattern were higher than matched and convergent pattern (matched: 5%; convergent: 3.8%, divergent 7.6%, *Figure 5E*).

## Morphologies analysis of tectal glutamatergic neurons

### Plasmids

To trace the morphology of tectal glutamatergic neurons, we designed three plasmids (*Figure 4B–C*). In the first plasmid, the Gal4FF cassette was placed at the start-codon site using a BAC recombination technique (*Suster et al., 2011*). Using this method, we created the Gal4FF BAC plasmids for 15 TF marker genes covering all 11 clusters of tectal glutamatergic neurons. Considering the promoter and other regulatory elements of higher expression genes may not be powerful to drive BAC plasmid expression, we selected six of them for their better performance in labeling tectal glutamatergic neuron morphologies based on two criterions: (1) These TFs are highly expressed and specific to individual tectal glutamatergic clusters based on scRNA-seq analysis. (2) Their BAC plasmids could reliably mark particular morphological subclasses (at least in four animals). The second BAC plasmid *vglut2a*:CRE was constructed using similar method with BAC (CH211-111D5). The CRE cassette was placed at the start-codon site of *vglut2a* (CH211-111D5) using BAC recombination technique. The third plasmid *uas*:loxp-stop-loxp-tdTomatocaax was constructed with the ClonExpressMultiS One Step Cloning Kit (Vazyme, C112-01/02) according to the standard protocol. We ligated three PCR fragments: loxp-stop-loxp, tdTomatocaax, and 10× uas backbone (*Figure 4C*). With these tree plasmids, *vglut2a* promoter-driven Cre-mediated excision, uas-Gal4 system activate tdTomatocaax expression as *Figure 4D* shown.

### Imaging

Embryos at 8 dpf were used for imaging. The fish were anesthetized with 0.04% MS222 (Sigma, A5040) and embedded in 1% agarose (Sigma, A0701) on imaging dish. Imaging were performed on Olympus FV1200 confocal using 30× oil-immersion objectives (Olympus, NA = 1.2). The image data were analysis with Image J (https://imagej.net/Simple_Neurite_Tracer). In total, we collected 574 single neuronal morphologies from 263 distinct fish (*Figure 4—source data 1* and *Figure 4—figure supplement 1G*). All neurons were categorized into seven morphological subclasses: bi-stratified (I, n = 36), mono-stratified (II, n = 71), non-stratified (III, n = 318), necklace-like (IV, n = 76), cross-hemispheric (V, n = 15), ascending (VI, n = 18), and descending (VII, n = 40) (*Figure 4D*).

### The analysis of TFs and tectal morphological subclasses

To calculate the expression of six TFs (*en2b*, *irx1a*, *zbtb18*, *foxb1a*, *bhlhe22*, and *zic1*) with respect to 11 tectal glutamatergic clusters, we created a binary expression matrix consisting of the gene expression (ON: with expression; OFF: without expression) and cluster identity (*Li et al., 2017*). We marked TFs that have expression in each of 11 clusters with cutoff of 5% cells or have averaged UMI of each TF-expressed cell UMI = 2 as ON, shown with black box. The rests were marked as OFF, shown with white box (*Figure 4—source data 1* and *Figure 4—figure supplement 1E-F*). All six TFs marked

multiple transcriptome-based subclasses in a combinatorial manner (*Figure 4—figure supplement 1F*).

To calculate the expression of six TFs with respect to seven tectal glutamatergic morphological subclasses, we also created a binary expression matrix consisting of the gene expression (ON: with expression; OFF: without expression) and morphological identity. We marked genes labeling one morphological subclass at least four times in individual four fish as ON, shown with black box. The rests were marked as OFF, shown with white box. All six TFs marked multiple morphological subclasses in a combinatorial manner (*Figure 4E*).

These two binary matrices were used to test the correlation between transcriptome-based clusters and morphological subclasses of tectal glutamatergic neurons with NNLS. In total, 6 out of 11 transcriptomic clusters exhibited strong correlation with at least one morphological subclass (*Figure 4—figure supplement 1H*).

## Data availability

scRNA-seq data has been deposited on BIG (CRA002361): https://ngdc.cncb.ac.cn/gsa/browse/CRA002361.

## Code availability

All the code for the data analysis is available.

## Acknowledgements

We thank Dr Muming Poo for the discussion and manuscript editing. We thank facilities of the Institute of Neuroscience: Dr Min Zhang (MZ) and Zhenning Zhou (ZZ) from Molecular and Cellular Biology Core Facility for the assistance of scRNA-seq; FACS facility Haiyan Wu (HW); the optical imaging facility Qian Hu (QH), Yumei Zhang (YZ), and Yonghong Wang (YW); the bioinformatics core facility and the facility of mapping brain-wide mesoscale connectome. This research was funded by grants from the Shanghai basic research field Project (Grant No.18JC1410100), Shanghai Municipal Science and Technology Major Project (Grant No. 2018SHZDZX05), The National Key Research and Development Program of China (Grant No. 2020YFA0112700), Strategic Priority Research Program of Chinese Academy of Science(Grant No. XDB32000000), National Natural Science Foundation of China (grant No. 31871035), State Key Laboratory of Neuroscience.

## Additional information

### Funding

| Funder | Grant reference number | Author |
| --- | --- | --- |
| Shanghai basic research field Project | 18JC1410100 | Jiulin Du<br>Jun Yan<br>Jie He |
| Shanghai Municipal Science and Technology Major Project | 2018SHZDZX05 | Jiulin Du<br>Jun Yan<br>Jie He |
| The National Key Research and Development Program of China | 2020YFA0112700 | Jie He |
| Strategic Priority Research Program of Chinese Academy of Science | XDB32000000 | Jiulin Du<br>Jun Yan<br>Jie He |
| National Natural Science Foundation of China | 31871035 | Jie He |
| State Key Laboratory of Neuroscience | | Jiulin Du<br>Jun Yan<br>Jie He |

| Funder | Grant reference number | Author |
|--------|------------------------|--------|

The funders had no role in study design, data collection and interpretation, or the decision to submit the work for publication.

## Author contributions

Hui Zhang, Conceptualization, Data curation, Formal analysis, Investigation, Methodology, Validation, Visualization, Writing – original draft; Haifang Wang, Formal analysis, Methodology; Xiaoyu Shen, Data curation, Formal analysis, Investigation, Methodology, Validation; Xinling Jia, Shuguang Yu, Xiaoying Qiu, Yufan Wang, Data curation; Jiulin Du, Jun Yan, Funding acquisition, Project administration, Supervision, Writing – review and editing; Jie He, Conceptualization, Funding acquisition, Project administration, Supervision, Writing – original draft, Writing – review and editing

## Author ORCIDs

Hui Zhang http://orcid.org/0000-0002-5300-5310
Xiaoyu Shen http://orcid.org/0000-0001-8868-7035
Shuguang Yu http://orcid.org/0000-0001-6640-5420
Jun Yan http://orcid.org/0000-0002-0405-0502
Jie He http://orcid.org/0000-0002-2539-2616

## Ethics

Animal procedures performed in this study were approved by the Animal Use Committee of Institute of Neuroscience, Chinese Academy of Sciences (NA-045-2019).

## Decision letter and Author response

Decision letter https://doi.org/10.7554/eLife.68224.sa1
Author response https://doi.org/10.7554/eLife.68224.sa2

# Additional files

## Supplementary files

• Transparent reporting form

## Data availability

Single cell RNA-seq data has been deposited on BIG (Genome Sequence Archive - CRA002361).

The following dataset was generated:

| Author(s) | Year | Dataset title | Dataset URL | Database and Identifier |
|-----------|------|---------------|-------------|--------------------------|
| Zhang H | 2021 | Single-cell RNA sequencing of zebrafish whole brain | https://ngdc.cncb.ac.cn/gsa/browse/CRA002361 | Genome Sequence Archive, CRA002361 |

The following previously published datasets were used:

| Author(s) | Year | Dataset title | Dataset URL | Database and Identifier |
|-----------|------|---------------|-------------|--------------------------|
| Raj B, Wagner DE, McKenna A, Pandey S, Klein AM, Shendure J, Gagnon JA, Schier AF | 2018 | GSE105010_fall.inDrops.RData.gz | https://www.ncbi.nlm.nih.gov/geo/query/acc.cgi?acc=GSE105010 | NCBI Gene Expression Omnibus, GSE105010 |

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
