## [Decision Letter]

**Decision letter after peer review:**

Thank you for submitting your article "The landscape of regulatory genes in brain-wide neuronal phenotypes of a vertebrate brain" for consideration by *eLife*. Your article has been reviewed by 3 peer reviewers, including Koichi Kawakami as the Reviewing Editor and Reviewer #1, and the evaluation has been overseen by a Reviewing Editor and Didier Stainier as the Senior Editor.

Essential revisions:

The reviewers recognized the authors performed enormous amounts of works and the data presented in the manuscript should be useful resources for neurobiologists. However the reviewers indicated the following major concerns which should be addressed by the authors.

1) The validity of clustering. The reviewers think the authors need control analysis for this.

2) For "divergent" and "convergent" classes, the hypothesis presented by the authors were not proven. To experimentally prove this is a bit too much for the present paper. Instead, the reviewers request more statistical data to support these ideas.

*Reviewer #1:*

In this manuscript, the authors performed single cell RNA-seq analysis of ~46,000 cells from the zebrafish larval brain and identified 68 clusters and mapped them on the brain regions. They found (1) region-specific markers, (2) correlation with cell types in juvenile fish, (3) 1099 effector genes out of 1402 genes used to define the clusters, (4) and 48 neuronal clusters and 20 non-neurons including glia. Then, they analyzed scRNA-seq using vmat (monoaminergic) and vachte (cholinergic) transgenic fish and (5) identified 22 and 14 neuromodulator clusters. (6) The neuromodulator clusters were further analyzed for neurotransmitter types and they revealed coexpression patterns.

In the hierarchical classification, they identified (7) 11 "sister clusters" that have similar expression profiles for effector genes, that had the same neurotransmitter types but did not show brain region preference. Then they examined TF profiles in neurotransmitter type neurons and identified 14 TF sister clusters. They found (8) one effector gene cluster that match with a TF cluster (matched) and 10 effector gene clusters that did not match with TF clusters (convergent) (these expressed the same neurotransmitters), and also found (9) neuromodulator clusters were well matched with TF profiles (matched). (10) They found the same TF clusters can express different effector genes also (divergent).

Then they aimed to see relationship between TF and morphology. They sorted tectal glut cells and identified 11 TF clusters. Then they made Bac-gal4 for 15 TFs and performed intersection using Cre-glut. (11) They analyzed 574 tectal neurons and found that TF could mark multiple morphology.

Since (in the divergent class) TF did not correlate with effector, they analyzed RNA binding proteins. (12) RNA-binding proteins were expressed differently in different neuron clusters.

Strength: The authors performed a comprehensive analysis of brain cells by single-cell RNA-seq. The scRNA-seq data presented here will be a good reference when the neuroscientists study a certain neuronal types, for instance searching for marker genes expressed in the neuronal clusters of their interest. Also, the amount of work is enormous.

Weakness: The contents of this paper are rather descriptive and poor in mechanistic aspects (causal relationship). Also I felt difficulty in following the manuscript since experiments were not described as hypothesis-driven.

1) The authors prepared cells for sc-RNA seq in different ways and this made the manuscript a bit confusing. Please clarify the purpose and reason for the to do so.

2) I think only "positive" data is the identification of the "matched" class. For "convergent" and "divergent" classes, many other explanations will be possible since there are no mechanistic analysis. For "convergent" class, examine if any TFs with weak expression had been overlooked. For "divergent" class, examine if the authors can find specific (classes) of RBP as real candidates.

3) As for the section describing the morphology, 11 clusters identified, and the 7 morphological classes seemed irrelevant. Also, relationship between 6TFs and the 7 morphological classes did not sound although the work for construction of 15 BACs should have to be enormous. Rewrite the section for readers to be understandable.

*Reviewer #2:*

In this study, the authors investigated how diverse neuronal types develop in the brain by using single-cell RNA sequencing methods.

The authors performed rigorous data collections for different brain regions, monoaminergic neurons, catecholaminergic neurons, and glutamatergic neurons. Such careful data collection for different types of neurons and brain regions has not been done for larval zebrafish. Their data will be a valuable resource for the neuroscience field.

They further found that the expression patterns of transcription factors are not necessarily predictive of the expression patterns effector genes that constitute the "terminal features" of neuronal cell types. Such predictiveness depends on the cell types. In neuromodulatory neurons, TF expression is predictive of effector gene expressions. On the contrary, in "neurotransmitter" neurons, which represent glutamatergic and GABAergic neurons, this relationship is loose and diverse. This finding, if confirmed, will advance our understanding of how diverse neuronal types develop in vertebrate brains.

The major weakness of this study is the lack of statistical controls in their analyses. I raise two examples here. First, the classification of "convergent types" and "divergent types" is solely based on hierarchical clustering analysis and its distance measures which are not validated by sub-sampling or by other statistical methods. Therefore, this analysis cannot rule out the possibility that such classification arises from large variations of gene expressions within the analyzed populations and that there are no such distinct populations.

The second example is their analysis of the expression of RNA-binding proteins. They show that the RBPs have differential expression in "divergent" cluster pairs. However, they do not show whether such differential expression is more prevalent among "divergent" cluster pairs than in other neuronal cluster pairs. If this is not the case, the differential expression of RBPs may not be the reason for the differential expression of effector genes.

Although the claim of this paper may be of broad interest in the neuroscience field, the above weaknesses significantly affect the reliability of the conclusion of this paper. Therefore, I recommend a revision of this manuscript for its publication in *eLife*.

1) The classification of "convergent" and "divergent" types in Figure 3 needs quantitative validation to exclude the possibility that such classification arises from large variations of gene expressions. This problem is unavoidable for analyses that solely rely on hierarchical clustering methods and their distance measures. Showing expression patterns of several example genes does not reinforce the conclusion, as it is always possible to find genes that show differential expression between any given cluster pairs. The result of neuromodulatory neurons only works as a partial control, as they are different neuronal types. This study needs to reinforce the validity of the classification by cross-validation of cluster distances among samples or by using unbiased statistical methods other than clustering.

2) The analysis of differential expression of RNA-binding proteins among "divergent" neuronal clusters needs statistical control. The authors need to show that RBPs in "divergent" pairs have more divergent expression patterns than in "convergent" pairs in an unbiased, quantitative way. Again, showing the example of few genes is not enough. Otherwise, RBPs cannot explain the divergence of effector gene expression from similar TF expression profiles.

3) The coloring schemes in figures are confusing. For example, I can see many color schemes in Figure 1. We only need two types of classification: (1) brain regions and (2) cell types. Figure 1a may not need colors/numberings and only need names for some of the clusters. Also, the classification presented in Figure 1c (Glu-GABA-P-R) and the one presented in Figure 1g (I- II, III, IV, V) are redundant. I understand that these different schemes serve different purposes, but I recommend unifying these classification schemes for clarity. This unification may need reordering of figure panels.

*Reviewer #3:*

In this manuscript, the authors performed single-cell RNA sequencing of >60,000 cells across the whole zebrafish brain with region- and molecular- identity. Using the acquired transcriptomes, the authors tried to deduce the regulation logic of neuronal diversification by comparing sister clusters in hierarchical clustering based on effector genes and regulatory TFs. The author showed that while TF similarity in modulatory neurons largely predict the similarity of their neuromodulator types, neurotransmitter types and their TF profiles usually do not agree. Further analysis of cell-type divergence from common TF regulators revealed an interesting differential enrichment of RNA binding proteins, which are potentially involved in post-transcriptional regulation of neuronal identity.

The transcriptomic data is comprehensive and of high quality, with cells covering the entire zebrafish brain, and with close-up analysis generated by new experiments to give higher discriminatory power. The data offers a valuable resource to the developmental neurobiology community. Some patterns (e.g. phenotypic convergence) echoes the findings in invertebrate nervous systems. Difference in regulatory logic of neurotransmitter vs. neuromodulator types, as well as the identification of post-transcriptional regulator genes are novel and interesting.

However, some caveats in data analysis may affect the reliability of the conclusions. The key claims in the study heavily relied on analysis of "sister clusters", i.e. clusters of cells with most similar TF or effector gene profiles. Yet not enough justification was given to the selection of such clusters or the focus only on the direct sibling clusters, and the fact that neurotransmitter and neuromodulator data were acquired differently adds complication to the interpretation of the result. Meanwhile, although the descriptive data in this study gives a detailed account of neuronal diversity, the lack of causal evidence and/or concrete mechanistic explanation between regulatory genes and terminal effectors rendered the conclusions a bit elusive -they tend to fell into providing interesting insight while failing to account for alternative explanations.

Overall, I think the claims are supported by the data for the most part, and with the addition of certain control and additional analyses, it enhances our understanding of vertebrate neuronal diversification as a thought-provoking descriptive study: Genes involved in convergent or divergent cell types identified in this study serve as curious candidate for follow-up investigation. Regulatory logic and mechanisms, when compared with similar studies in invertebrates, can help us to gain insights on the origin and evolution of the nervous system.

1. In the first and second section of Result, clusters of the transcriptome data were treated as the "smallest unit" for subsequent analysis. However, there's a lack of justification for the clustering criteria: i.e. how distinct the clusters are, and how robust the subsequent analysis result is if cell-type clustering is performed slightly differently. This is especially problematic because even the authors themselves have shown that given cleaner quality data (e.g. modulatory neurons in FAC sorted cells vs. whole brain), clustering partition could be different. Additional control analysis would be necessary to show clustering makes sense and robust to noise level in the data.

2. The authors compared the 8dpf brain transcriptomes with juvenile brain in Raj et al., 2018 and claimed that "are likely to represent the full cellular diversity of the mature zebrafish brain". The stretch is a bit far because they partitioned the data only using variable genes expressed in both datasets, i.e. differentially expressed genes involved in subsequent diversification were not accounted for. I think at best this analysis serves as a similarity claim rather than "full" cellular diversity.

3. Result Section 3: a major weakness in the design of analysis is the focus only on "sister clusters", which can be sensitive to your cluster partition and does not necessarily represent the real diversification event. To compare the landscape of TF and effector gene expression, there are many alternative methods accounting for the full spectrum of cell types rather than just the most similar sibling clusters. I would suggest two supplementary analyses: (1) A population-level statistical analysis to show the difference in regulatory logic across all cell types regardless of clustering, even in the less similar cells. (2) A test to ensure that the disagreement between TF and effectors does not disappear as we break clusters in neurotransmitter types into smaller sub types. As stated in point #1, it is necessary for the authors to demonstrate the difference in regulatory logic between transmitter and modulator types is not a result of different noise level in the data.

4. Result Section 4: the criteria for assigning TF expression to morphology class is again relying too much on binary classification. In particular, the authors considered a TF to be a marker for the morphology class "appeared for at least 4 times". The morphology classes are very different in sizes and this is an unfair comparison. Rather more quantitative metrics and vigorous statistical tests should be used to support this conclusion.

5. Result Section 5: in addition to transcriptome data, it would be nice if the authors can demonstrate some causal links between the RBP genes and the cell type regulation, or verify experimentally that they indeed encode neurons with different identity, morphology or functions. If additional experiments to demonstrate causal links are not possible, the authors should sufficiently account for alternative explanations for cell type divergence. Besides post-transcriptional regulation, there are a lot of other factors that could affect gene diversification, including early regulators that are transiently expressed in the embryo that primed fate selection, and external signaling factors in the neuron's environment. Such information was lost as only mature neurons are sequenced with very little spatial context.

[Editors' note: further revisions were suggested prior to acceptance, as described below.]

Thank you for resubmitting your work entitled "The landscape of regulatory genes in brain-wide neuronal phenotypes of a vertebrate brain" for further consideration by *eLife*. Your revised article has been evaluated by Didier Stainier (Senior Editor) and a Reviewing Editor.

The manuscript has been corrected but there are some remaining issues that need to be addressed, as outlined below:

The reviewers still think the data is a good resource, and the differentially expressed genes can be good candidates for future investigation. However, the reviewers think that cross-validation of their statistical measures that used to identify "matched", "convergent" and "divergent" types is necessary. They used two measures in parallel, hierarchical clustering, and population-level similarity, to identify these pairs. These measures can be made robust by repeating the sub-sampling of genes and average statistics over many sub-sampling calculations. The reviewers wonder how much of it remain true with sub-sampling test added. Or alternatively, the reviewers would suggest moving away from hierarchical clustering and instead re-identifying matched, convergent and divergent populations based on TF-based and effector-based distance measures, which is bootstrapped to ensure robustness. This will provide more interpretable, consistent results.

*Reviewer #1:*

I found that the authors performed Jaccard similarity index-based analysis to validate clustering and rewrote the text. However, these revealed some weakness of the manuscript. I have the following comments.

1. The paper lacks biological aspects. The authors identified sister clusters from different brain regions (p11, line 234). The authors should at least discuss the significance of the finding.

2. The authors separated IIa and IIb before classification using TF profiles. I am wondering why?

3. A paragraph (p12, lines 249-258), it is not clear how they performed the population-level statistical analysis. Explain more precisely.

4. Why did the hierarchical and population analyses make the difference? (p12) Explain. Also, I do not understand the meaning of making overlaps of these.

5. p13, line 277. I think the authors examined both the TF and effector profiles, correct?

6. p13, Again the paper lacks biological aspects. Why did the neurotransmitter-type and neuromodulator-type make difference? Explain the idea in the discussion.

7. p14, lines 287-297, on the other hand… this part seems redundant with the previous parts and, to me, was difficult to read. Please clarify.

8. p14-15. I admire their efforts to make many BACs. They mentioned this may support "divergent pattern"(p17, line 358), but the reason why they thought so is very weak. They need to show relevance between morphology and effector type somehow experimentally.

9. p16, line 340. They did not describe how they picked up 6 TFs.

10. Figure3—figure supplement 2D and 2F. Figure3—figure supplement 4B. I could not tell how to read these data. In addition, in all cases, the effector-based distances are small (between 0-0.1) and TF-based distances are large (between 0.1-0.4). I don't see much difference between these populations.

*Reviewer #2:*

I have to say the authors did not address my concerns at all.

My first concern was about the validity of "matched", "convergent" and "divergent" classifications from hierarchical clusterings that are based on effector genes and TF genes in Figures 3 and 5. As an answer, the authors calculated Jaccard similarity indices for clusterings using all genes in Figure 1 and Figure 2. These added analyses were completely unrelated to the clustering of "matched", "convergent" and "divergent." Therefore, I have to say the authors did not address the issue.

My second concern was about the validity of their claim about the differential expression of RBPs in "divergent cell population". They claim that the RBPs have more divergent expressions in "divergent cell pairs". However, it is hard to interpret the statistics presented in Figure 5E. The indication of the index [(number of genes that are solely divergent in group A) / (number of all genes that are divergent in group A)] is unclear.

Moreover, this analysis does not indicate any causality between the divergence of RBPs and the divergence of effector genes. In principle, the more heterologous the population is, we would see more divergence of RBPs and effector genes. These factors are not independent of each other. I was expecting to see a more careful statistical approach that takes these considerations into account. Therefore, I have to say the authors did not address the issue.

*Reviewer #3:*

In the revised manuscript, the authors mainly added three additional control analyses that increased the credibility of the conclusion, including:

1) Robustness analysis of the clustering result using Jaccard similarity of subsampled data.

2) Analysis of the matched/convergent/divergent patterns of the glutamatergic/GABAergic neurons based on relaxed criteria that include not only the terminal sister clusters, but also non-sibling clusters with relatively close relationships.

3) Control analysis to show the enrichment of differential RBP expression in divergent patterns.

The control analyses largely addressed my concerns to a large extent, although they made the dramatic contrast between the regulatory logic of neurotransmitter and neuromodulator data weaker. The identified convergent/divergent events are now more convincing and allow for further exploration of the phenotype in the future.

However, I cannot help but noticed many more changes have been made without being pointed out in the rebuttal letter, including (1) exclusion of the data from the cholinergic neurons altogether, and (2) slightly different analysis result with seemingly identical input data (e.g. comparing Figure 3—figure supplement 2 in the new version vs. Figure 3 in the original manuscript). The authors should explain why such changes were made and ensure there is no selective report of the data.

Finally, some suggestions/questions to help achieve a better presentation of the data.

1) The current "population-level" analysis is rather just a relaxed definition of the terminal sister clusters, but there are many other metrics to quantify the similarity between the TF-based and effector-based clustering result over the entire dataset. For hierarchical clustering this can be editing distance between the two trees, or more generally, similarity between organization of TF and effector gene distribution across all the clusters. This will not identify specific matched, convergent or divergent patterns, but would be an unbiased measurement of how TF and effector landscape agree with each other.

2) Figure 3: similar color regime was used to represent forebrain/OT/sub OT and matched/convergent/divergent patterns. This is very confusing. I would suggest weakening brain region visualization (e.g. changing it to a less pronounced representation or getting rid of it altogether -see comment #3) and focus on the regulatory logic. It is also recommended to color the pattern names in 3B accordingly.

3) Figure 3C: The number of matched patterns on the Venn diagram does not match the number shown in Figure 3F. Also, Venn diagram is not the best way to show that the conclusion in sister cluster- and population-statistics are similar. [Minor correction: the number of selection should be Combination C(39, 2) rather than Permutation A(39, 2)].

4) Throughout the manuscript, the brain region identities are shown next to the clusters in every figure, yet very little conclusion was made about the relationship between transcriptome and brain region identity. The authors are advised to summarize the major findings of data feature related to brain region identities (no correlation can also be a finding), and simplify the color use of brain region when the color does not convey meaningful information.

[Editors' note: further revisions were suggested prior to acceptance, as described below.]

Thank you for resubmitting your work entitled "The landscape of regulatory genes in brain-wide neuronal phenotypes of a vertebrate brain" for further consideration by *eLife*. Your revised article has been evaluated by Didier Stainier (Senior Editor), a Reviewing Editor, and the original reviewers.

The manuscript has been improved but there are some remaining issues that need to be addressed, as outlined below:

1) Although for neurotransmitter-type, "sister cluster analysis and the population-level statistical analysis " are carried out. However, for neuromodulator-type, only sister cluster analysis was done. It is fair to perform population-level analysis for neuromodulator-type.

2) Line 265: largely recapitulated. I do not agree with this. Rather, explain why the results were not consistent.

3) Line 226: The TF Regulatory Landscape in Whole-brain Neuronal Phenotypes

This section is a bit too long. The section should be separated, for example, for neurotransmitter-type and neuromodulator-type.

4) Line 395: results supported "divergent" and "convergent" patterns, because divergent pattern indicated the link of same TFs to different neuron phenotypes, while convergent pattern indicated the link of different TFs to similar neuron phenotypes.

This is too much speculation. The relationship between the morphology and effector gene profiles are unclear. I think it is fair to say that morphology does not directly correlate with TF profiles.

*Reviewer #2:*

I can say the authors addressed my concerns on statistical analyses after this round of revision, shrugging off unnecessary parts and exposing the most interesting part of this study. In addition, they included a new analysis in Figure 5 which indicates potential causality between the differential expression of RBPs and differential expression of effector genes.

There are still concerns about the validity of classification (lower 10% of gene distance compared to shuffled pairs, how do you justify the threshold?). However, this justification will not be so straightforward because it will be prone to the intrinsic statistics of TF expression, effector gene expression, and the sensitivity of the sequencing technique. It is obvious this is the best authors can do. I advise authors to acknowledge such limitations in the discussion.

*Reviewer #3:*

Unfortunately, the authors misunderstood my request for adding Tree Distance analysis. An important claim of the paper was that glutamatergic/GABAergic clusters show different TF and terminal profile patterns ("convergent/divergent"), whereas neuromodulator type clusters predominantly expressed the same TF profiles ("matched"). However, supplementary analysis was only shown for the glutamatergic/GABAergic types but not the modulator types, and the main figure that supports this conclusion (Figure 3E) still uses sister clusters from hierarchical clustering for classification of neuromodulator types, which I found to be an unfair comparison. Also, the gene subsampling test of the robustness of matched/divergent/convergent pairs was only performed on similar pairs defined by population-statistics. It was not clear to me why the authors still adhered to the "sister cluster" definition (or an INTERSECTION of this strategy with the population-statistics), for all subsequent conclusions. I suggested tree distance as a global measurement of whether the glutamatergic/GABAergic types and modulator types truly have different regulatory logic at the population level, rather only at the terminal sisters.

---

## [Author Response]

Essential revisions:The reviewers recognized the authors performed enormous amounts of works and the data presented in the manuscript should be useful resources for neurobiologists. However the reviewers indicated the following major concerns which should be addressed by the authors.1) The validity of clustering. The reviewers think the authors need control analysis for this.

We have introduced Jaccard similarity index-based analysis by subsampling to validate our clusters (Figure 1—figure supplement 1, whole-brain clusters; Figure 2—figure supplement 1A, *vmat*^+^ neuromodulator clusters; Figure 4—figure supplement 1D, tectal glutamatergic clusters) (Tang et al., 2020).

Specifically, to validate 68 clusters derived from whole-brain samples, we introduced R package “scclusteval”. 80 % of total cells were subsampled and re-clustered. The Jaccard index was then used to measure the similarity of clusters derived from subsampled cells and total cells. The mean/median of Jaccard index after 20-times repeats was used to evaluate the robustness of original 68 clusters. One cluster with a mean/median of Jaccard index above 0.6 is considered as a stable cluster as described previously (Zumel N. , 2014).

2) For "divergent" and "convergent" classes, the hypothesis presented by the authors were not proven. To experimentally prove this is a bit too much for the present paper. Instead, the reviewers request more statistical data to support these ideas.

Thanks for the constructive suggestion. In the revised manuscript, to compare the landscape of TF and effector gene expression accounting for the full spectrum of cell types rather than just the most similar sister clusters, we also performed population-level statistical analysis to support the idea of “divergent” and “convergent” patterns (Figure 3A-B, Figure 3—figure supplement 2D-G, and Figure 3—figure supplement 4B-C).

Reviewer #1:In this manuscript, the authors performed single cell RNA-seq analysis of ~46,000 cells from the zebrafish larval brain and identified 68 clusters and mapped them on the brain regions. They found (1) region-specific markers, (2) correlation with cell types in juvenile fish, (3) 1099 effector genes out of 1402 genes used to define the clusters, (4) and 48 neuronal clusters and 20 non-neurons including glia. Then, they analyzed scRNA-seq using vmat (monoaminergic) and vachte (cholinergic) transgenic fish and (5) identified 22 and 14 neuromodulator clusters. (6) The neuromodulator clusters were further analyzed for neurotransmitter types and they revealed coexpression patterns.In the hierarchical classification, they identified (7) 11 "sister clusters" that have similar expression profiles for effector genes, that had the same neurotransmitter types but did not show brain region preference. Then they examined TF profiles in neurotransmitter type neurons and identified 14 TF sister clusters. They found (8) one effector gene cluster that match with a TF cluster (matched) and 10 effector gene clusters that did not match with TF clusters (convergent) (these expressed the same neurotransmitters), and also found (9) neuromodulator clusters were well matched with TF profiles (matched). (10) They found the same TF clusters can express different effector genes also (divergent).Then they aimed to see relationship between TF and morphology. They sorted tectal glut cells and identified 11 TF clusters. Then they made Bac-gal4 for 15 TFs and performed intersection using Cre-glut. (11) They analyzed 574 tectal neurons and found that TF could mark multiple morphology.Since (in the divergent class) TF did not correlate with effector, they analyzed RNA binding proteins. (12) RNA-binding proteins were expressed differently in different neuron clusters.Strength: The authors performed a comprehensive analysis of brain cells by single-cell RNA-seq. The scRNA-seq data presented here will be a good reference when the neuroscientists study a certain neuronal types, for instance searching for marker genes expressed in the neuronal clusters of their interest. Also, the amount of work is enormous.Weakness: The contents of this paper are rather descriptive and poor in mechanistic aspects (causal relationship). Also, I felt difficulty in following the manuscript since experiments were not described as hypothesis-driven.1) The authors prepared cells for sc-RNA seq in different ways and this made the manuscript a bit confusing. Please clarify the purpose and reason for the to do so.

Sorry for the confusion. We have clarified the reasons of preparing cells for scRNAseq using different methods in the revision.

Specifically, we used three methods to isolate cells for scRNAseq: a. from whole-brain samples, b. from samples of particular brain regions, and c. from transgenic fishlines with labeling of neurons expressing given neurotransmitters or neuromodulators.

To validate our data at whole-brain level, we took advantage of whole-brain samples to get a whole picture of transcriptome in larval zebrafish brain. Besides, for quantifying the ratio of different neurotransmitter/modulator-type neurons at whole-brain level (Figure 1—figure supplement 2C-D), we used cells isolated from whole-brain samples.

For studying regional identities, we dissected anatomically distinct brain samples (Figure 1—figure supplement 2B).

Since we found few neuromodulator clusters (n=3) in ~46,000 cells from samples of whole brains and different brain regions, we sorted cells from transgenic fishlines that specifically marked neuromodulators to analyze enriched neuromodulator populations (Figure 2B).

For studying the relationship between transcriptomic clusters and morphological subclasses for tectal glutamatergic neurons, we need finer-defined transcriptomic glutamatergic clusters. Thus, we further verify tectal glutamatergic clusters by performing additional scRNAseq analysis on cells isolated from Tg (*vgluta2a*:loxP-DsRed-loxp-GFP, Figure 4A)

2) I think only "positive" data is the identification of the "matched" class. For "convergent" and "divergent" classes, many other explanations will be possible since there are no mechanistic analysis. For "convergent" class, examine if any TFs with weak expression had been overlooked. For "divergent" class, examine if the authors can find specific (classes) of RBP as real candidates.

For identifying “convergent” classes, we used variable TFs (n = 283) based on all 68 clusters, that is some were expressed at higher levels and others were expressed at lower levels for a given cluster. Thus, our analysis does not particularly overlooked TFs with weak expression.

For two clusters with “divergent pattern”, we found that RBPs could be specifically expressed in one but not the other (Figure 5B). Within 121 RBPs that distinguish paired clusters of three patterns, 38 % RBPs are specific to divergent pattern, while 6.6 % RBPs are specific to either convergent or matched patterns (Figure 5D-E).

3) As for the section describing the morphology, 11 clusters identified, and the 7 morphological classes seemed irrelevant. Also, relationship between 6TFs and the 7 morphological classes did not sound although the work for construction of 15 BACs should have to be enormous. Rewrite the section for readers to be understandable.

Sorry for the confusion. We have further clarified the purpose and logic of this part and a newly-analyzed correlation data in the revised manuscript.

Since we described “convergent pattern” (different TFs could mark neuron clusters with similar effector-gene profiles) and “divergent pattern” (single TFs could mark neuron clusters with different effector-gene profiles) in Figure 3, we then decided to provide the experimental evidence to support the presence of both phenotypes by analyzing the relationship between TFs and different morphological neuron subclasses in the tectum.

Specifically, we identified 11 morphological subclasses of tectal glutamatergic neurons, and 6 TFs that showed highly variable expressions among these 11 clusters. Then by creating BACs for each of these 6 TFs, we showed that each of the 6 TFs were mostly used by multiple morphological neuron subclasses in a combinatorial manner, which supported “divergent pattern”; meanwhile, each of 7 morphological subclasses could be marked by multiple TFs, providing the evidence for “convergent pattern” as Figure 4E.

Reviewer #2:In this study, the authors investigated how diverse neuronal types develop in the brain by using single-cell RNA sequencing methods.The authors performed rigorous data collections for different brain regions, monoaminergic neurons, catecholaminergic neurons, and glutamatergic neurons. Such careful data collection for different types of neurons and brain regions has not been done for larval zebrafish. Their data will be a valuable resource for the neuroscience field.They further found that the expression patterns of transcription factors are not necessarily predictive of the expression patterns effector genes that constitute the "terminal features" of neuronal cell types. Such predictiveness depends on the cell types. In neuromodulatory neurons, TF expression is predictive of effector gene expressions. On the contrary, in "neurotransmitter" neurons, which represent glutamatergic and GABAergic neurons, this relationship is loose and diverse. This finding, if confirmed, will advance our understanding of how diverse neuronal types develop in vertebrate brains.The major weakness of this study is the lack of statistical controls in their analyses. I raise two examples here. First, the classification of "convergent types" and "divergent types" is solely based on hierarchical clustering analysis and its distance measures which are not validated by sub-sampling or by other statistical methods. Therefore, this analysis cannot rule out the possibility that such classification arises from large variations of gene expressions within the analyzed populations and that there are no such distinct populations.

Thanks for the question and great suggestions. In the revised manuscript, we have verified the neuron classification by sub-sampling, and the Jaccard similarity index (Tang et al., 2020) were introduced to further validate our “convergence pattern” and “divergent pattern”.

The second example is their analysis of the expression of RNA-binding proteins. They show that the RBPs have differential expression in "divergent" cluster pairs. However, they do not show whether such differential expression is more prevalent among "divergent" cluster pairs than in other neuronal cluster pairs. If this is not the case, the differential expression of RBPs may not be the reason for the differential expression of effector genes.

Thanks for the question and great suggestions. We have included a new analysis in the revision, which shows that such differential expression of RBPs are more prevalent among “divergent” cluster pairs than in other pairs in terms of gene numbers and differential expression levels (Figure 5D-E, Figure 5—figure supplement 1C-E).

Although the claim of this paper may be of broad interest in the neuroscience field, the above weaknesses significantly affect the reliability of the conclusion of this paper. Therefore, I recommend a revision of this manuscript for its publication in eLife.1) The classification of "convergent" and "divergent" types in Figure 3 needs quantitative validation to exclude the possibility that such classification arises from large variations of gene expressions. This problem is unavoidable for analyses that solely rely on hierarchical clustering methods and their distance measures. Showing expression patterns of several example genes does not reinforce the conclusion, as it is always possible to find genes that show differential expression between any given cluster pairs. The result of neuromodulatory neurons only works as a partial control, as they are different neuronal types. This study needs to reinforce the validity of the classification by cross-validation of cluster distances among samples or by using unbiased statistical methods other than clustering.

Thanks for the question and great suggestions. We have performed Jaccard similarity index-based analysis by sub-sampling (Tang et al., 2020) to further validate the classification (as shown in Figure 1—figure supplement 1B-C, Figure 2—figure supplement 1A).

2) The analysis of differential expression of RNA-binding proteins among "divergent" neuronal clusters needs statistical control. The authors need to show that RBPs in "divergent" pairs have more divergent expression patterns than in "convergent" pairs in an unbiased, quantitative way. Again, showing the example of few genes is not enough. Otherwise, RBPs cannot explain the divergence of effector gene expression from similar TF expression profiles.

Thanks for the question and great suggestions. We have included a new analysis in the revision, which shows that such differential expression of RBPs are more prevalent among pairs with “divergent” cluster pairs than in other patterns in terms of gene numbers and differential expression levels (as shown in Figure 5D-E, and Figure 5—figure supplement 1C-E).

3) The coloring schemes in figures are confusing. For example, I can see many color schemes in Figure 1. We only need two types of classification: (1) brain regions and (2) cell types. Figure 1a may not need colors/numberings and only need names for some of the clusters. Also, the classification presented in Figure 1c (Glu-GABA-P-R) and the one presented in Figure 1g (I- II, III, IV , V ) are redundant. I understand that these different schemes serve different purposes, but I recommend unifying these classification schemes for clarity. This unification may need reordering of figure panels.

Thanks for the great suggestions. We have updated the color schemes in Figure 1A in the revision using major cell type in brain. Also, we have updated figure panels in the Figure 1F.

Reviewer #3:In this manuscript, the authors performed single-cell RNA sequencing of >60,000 cells across the whole zebrafish brain with region- and molecular- identity. Using the acquired transcriptomes, the authors tried to deduce the regulation logic of neuronal diversification by comparing sister clusters in hierarchical clustering based on effector genes and regulatory TFs. The author showed that while TF similarity in modulatory neurons largely predict the similarity of their neuromodulator types, neurotransmitter types and their TF profiles usually do not agree. Further analysis of cell-type divergence from common TF regulators revealed an interesting differential enrichment of RNA binding proteins, which are potentially involved in post-transcriptional regulation of neuronal identity.The transcriptomic data is comprehensive and of high quality, with cells covering the entire zebrafish brain, and with close-up analysis generated by new experiments to give higher discriminatory power. The data offers a valuable resource to the developmental neurobiology community. Some patterns (e.g. phenotypic convergence) echoes the findings in invertebrate nervous systems. Difference in regulatory logic of neurotransmitter vs. neuromodulator types, as well as the identification of post-transcriptional regulator genes are novel and interesting.However, some caveats in data analysis may affect the reliability of the conclusions. The key claims in the study heavily relied on analysis of "sister clusters", i.e. clusters of cells with most similar TF or effector gene profiles. Yet not enough justification was given to the selection of such clusters or the focus only on the direct sibling clusters, and the fact that neurotransmitter and neuromodulator data were acquired differently adds complication to the interpretation of the result. Meanwhile, although the descriptive data in this study gives a detailed account of neuronal diversity, the lack of causal evidence and/or concrete mechanistic explanation between regulatory genes and terminal effectors rendered the conclusions a bit elusive -they tend to fell into providing interesting insight while failing to account for alternative explanations.Overall, I think the claims are supported by the data for the most part, and with the addition of certain control and additional analyses, it enhances our understanding of vertebrate neuronal diversification as a thought-provoking descriptive study: Genes involved in convergent or divergent cell types identified in this study serve as curious candidate for follow-up investigation. Regulatory logic and mechanisms, when compared with similar studies in invertebrates, can help us to gain insights on the origin and evolution of the nervous system.1. In the first and second section of Result, clusters of the transcriptome data were treated as the "smallest unit" for subsequent analysis. However, there's a lack of justification for the clustering criteria: i.e. how distinct the clusters are, and how robust the subsequent analysis result is if cell-type clustering is performed slightly differently. This is especially problematic because even the authors themselves have shown that given cleaner quality data (e.g. modulatory neurons in FAC sorted cells vs. whole brain), clustering partition could be different. Additional control analysis would be necessary to show clustering makes sense and robust to noise level in the data.

Thanks for the questions and great suggestions. In the revision, we further performed Jaccard similarity index-based analysis by sub-sampling, showing that sub-sampling of cells could give rise to the similar clustering partition, supporting the robustness of our clustering (Figure 1—figure supplement 1E, whole-brain clusters; Figure 2—figure supplement 1A, *vmat*^+^ neuromodulator clusters; Figure 4—figure supplement 1D, tectal glutamatergic clusters)(Tang et al., 2020).

Specifically, to validate 68 clusters derived from whole-brain samples, we introduced R package “scclusteval”. 80 % of total cells were subsampled and re-clustered. The jaccard index was then used to measure the similarity of clusters derived from subsampled cells and total cells. The mean/median of Jaccard index after 20-times repeats was used to evaluate the robustness of original 68 clusters. One cluster with a mean/median Jaccard index above 0.6 is considered as a stable cluster as described previously (Zumel N. , 2014).

2. The authors compared the 8dpf brain transcriptomes with juvenile brain in Raj et al., 2018 and claimed that "are likely to represent the full cellular diversity of the mature zebrafish brain". The stretch is a bit far because they partitioned the data only using variable genes expressed in both datasets, i.e. differentially expressed genes involved in subsequent diversification were not accounted for. I think at best this analysis serves as a similarity claim rather than "full" cellular diversity.

Thanks for this constructive suggestion. We have updated our conclusion to “our analysis indicated that the brain at 8 dpf mostly represented cellular diversity in the juvenile brain.”

3. Result Section 3: a major weakness in the design of analysis is the focus only on "sister clusters", which can be sensitive to your cluster partition and does not necessarily represent the real diversification event. To compare the landscape of TF and effector gene expression, there are many alternative methods accounting for the full spectrum of cell types rather than just the most similar sibling clusters. I would suggest two supplementary analyses: 1) A population-level statistical analysis to show the difference in regulatory logic across all cell types regardless of clustering, even in the less similar cells. 2) A test to ensure that the disagreement between TF and effectors does not disappear as we break clusters in neurotransmitter types into smaller sub types. As stated in point #1, it is necessary for the authors to demonstrate the difference in regulatory logic between transmitter and modulator types is not a result of different noise level in the data.

Thanks for raising this concern and the constructive suggestions. Thanks for the constructive suggestion. In the revised manuscript, we performed Jaccard similarity index-based analysis by sub-sampling, showing that sub-sampling of cells could give rise to the similar clustering partition, supporting the robustness of our clustering (Tang et al., 2020). Then to compare the landscape of TF and effector gene expression accounting for the full spectrum of cell types rather than just the most similar sister clusters, we also performed population-level statistical analysis to support the idea of “divergent” and “convergent” classes (Figure 3A, Figure 3—figure supplement 2D-G, Figure 3—figure supplement 4B-C).

4. Result Section 4: the criteria for assigning TF expression to morphology class is again relying too much on binary classification. In particular, the authors considered a TF to be a marker for the morphology class "appeared for at least 4 times". The morphology classes are very different in sizes and this is an unfair comparison. Rather more quantitative metrics and vigorous statistical tests should be used to support this conclusion.

Thanks for raising this concern. We have made a further clarification on this issue in the revision. During the analysis, we appreciated the enormous variability in the morphology of tectal glutamatergic neurons in the finer structures. By the limited number of neurons we analyzed (n=574), we were unlikely to define morphological subclasses using a full morphological description. Instead, we used the criterion based on major morphological features, including stratification, soma position, and projection patterns, to define the morphological subclasses in the current study. In addition, non-stratified, mono-stratified, and bi-stratified subclasses we defined were also used previously for tectal neurons (Robles et al., 2011). We have discussed these caveats in morphology classification in the revision. Furthermore, we have made the further clarification in the revision that our conclusion on the relationship between the TFs and morphology was based on the criteria using major morphological features mentioned above.

5. Result Section 5: in addition to transcriptome data, it would be nice if the authors can demonstrate some causal links between the RBP genes and the cell type regulation, or verify experimentally that they indeed encode neurons with different identity, morphology or functions. If additional experiments to demonstrate causal links are not possible, the authors should sufficiently account for alternative explanations for cell type divergence. Besides post-transcriptional regulation, there are a lot of other factors that could affect gene diversification, including early regulators that are transiently expressed in the embryo that primed fate selection, and external signaling factors in the neuron's environment. Such information was lost as only mature neurons are sequenced with very little spatial context.

Thanks for the question and great suggestions. We have included a new analysis in the revision, which further shows that such differential expression of RBPs in “divergent” clusters are more prevalent than those from other pairs in terms of gene numbers and differential expression levels in Figure 5D-E and Figure 5—figure supplement 1C-E. Furthermore, we included an additional discussion on the alternative explanations for neuron type divergence as suggested.

[Editors' note: further revisions were suggested prior to acceptance, as described below.]

The reviewers still think the data is a good resource, and the differentially expressed genes can be good candidates for future investigation. However, the reviewers think that cross-validation of their statistical measures that used to identify "matched", "convergent" and "divergent" types is necessary. They used two measures in parallel, hierarchical clustering, and population-level similarity, to identify these pairs. These measures can be made robust by repeating the sub-sampling of genes and average statistics over many sub-sampling calculations. The reviewers wonder how much of it remain true with sub-sampling test added. Or alternatively, the reviewers would suggest moving away from hierarchical clustering and instead re-identifying matched, convergent and divergent populations based on TF-based and effector-based distance measures, which is bootstrapped to ensure robustness. This will provide more interpretable, consistent results.

Thank you for the suggestions.

Besides hierarchical clustering, we performed the population-level statistical analysis in our last revision based on TF-based and effector-based distance measures to re-identify “matched”, “convergent”, and “divergent” pairs. These re-identified pairs mostly recapitulate those identified using hierarchical clustering (Figure 3B), indicating the robustness of three patterns.

In the current revision, we have further performed the population-level statistical analysis by sub-sampling of genes (80% of total either TFs or effector genes) and average statistics over 20 times to re-identify “matched”, “convergent”, and “divergent” pairs. These re-identified pairs completely recapitulated those identified using the statistical analysis based on total genes (Figure 3—figure supplemental 4E), indicating the robustness of three patterns.

Reviewer #1:I found that the authors performed Jaccard similarity index-based analysis to validate clustering and rewrote the text. However, these revealed some weakness of the manuscript. I have the following comments.1. The paper lacks biological aspects. The authors identified sister clusters from different brain regions (p11, line 234). The authors should at least discuss the significance of the finding.

Thanks for your suggestions. We have discussed the significance of different brain-region origins of sister clusters in the revision (p24, line 513-527).

First, since sister clusters in the same brain region may result from over-clustering, sister clusters from different brain regions showed the robustness of sister clusters.

Second, because neuronal phenotypes from the same region could exhibit common regional identities, it was surprising to observe that such a higher proportion of sister clusters with similar transcriptomic phenotypes could arise from two different brain regions (Figure 3—figure supplementary 2A). Putting this finding into the context of neuron type evolution raises the possibility that different brain regions independently give rise to similar neurotransmitter phenotypes through different TF programs. Alternatively, as brain regions are functionally diversified during the evolution, these highly similar neuronal clusters of different brain regions possibly derive from ancient building blocks, which become divergent through the evolutionary acquisition of different combinatorial TF codes.

2. The authors separated IIa and IIb before classification using TF profiles. I am wondering why?

Thanks for the question.

Our initial transcriptome-based classification showed that highly variable genes among all single cells were primarily effector genes (Figure 1—figure supplementary 2G), suggesting its critical role in brain cell classification. Previous studies also reported that effector genes better described the phenotype of neurons (Paul, Crow et al. 2017, Hodge, Bakken et al. 2019). Thus, to better classify neuron phenotypes, we first separated clusters using effector gene-based classification and identified IIa and IIb branch as glutamatergic and GABAergic neurons according to their marker genes (Figure 1-Source Data 5). To further address how regulatory genes determine effector gene-based neuron paired clusters, we introduced the classification using TFs and compared TF-based and effector gene-based classification to identify three patterns (Figure 3).

3. A paragraph (p12, lines 249-258), it is not clear how they performed the population-level statistical analysis. Explain more precisely.

Sorry for the confusion. We further clarified the population-level statistical analysis in the revision (p12-13, line 255-266).

For all glutamatergic/GABAergic neuronal clusters (n=39), we calculated the distances between every two clusters (C392) based on either effector gene profiles or TF profiles, then defined the pairs, which had the lowest 10% distances after ranking, as similar pair clusters (Figure 3A). Then, we defined pair clusters that were similar in both TF and effector gene profiles as “matched pattern”, those pair clusters that were similar in effector gene profiles but not in TF profiles as “convergent pattern”, and those pair clusters that were similar in TF profiles but not in effector gene profiles as “divergent pattern” (Figure 3B).

More detailed description has been enlisted in Materials and methods.

4. Why did the hierarchical and population analyses make the difference? (p12) Explain. Also, I do not understand the meaning of making overlaps of these.

Sorry for the confusion. We have clarified this issue in the revision (p13, line 275-277).

We were also aware of the weakness of hierarchical clustering: initial seeds, the order of data, and outlier data points have influences on the final hierarchical tree structure. Specifically, once a neuronal cluster has been assigned as a sister cluster with another cluster, it could no longer be paired with others. Therefore, we introduced a population-level statistical analysis to re-identify similar pair clusters, in which we calculated the distances between every two clusters (C392) based on either effector gene profiles or TF profiles, and defined the pairs, which have the lowest 10% distances after ranking, as similar pair clusters (Figure 3A; Details in Materials and methods). This analysis could identify similar pair clusters in a more unbiased manner.

Thus, we combined two methods to consolidate “matched”, “convergent”, and “divergent” types independently. The results showed that neuron pairs of three patterns were mostly similar using two strategies (Figure 3B). We overlap these paired clusters to make a better understanding of each cluster.

5. p13, line 277. I think the authors examined both the TF and effector profiles, correct?

Sorry for the confusion. Yes, we examined both TFs and effector profiles in Figure 3—figure supplementary 3A-B. We update this part in the revision (p14, line 291-292).

6. p13, Again the paper lacks biological aspects. Why did the neurotransmitter-type and neuromodulator-type make difference? Explain the idea in the discussion.

Thank you for the suggestions. We have discussed this idea in the revised manuscript (p14, line 296-307).

Our analysis showed that neuromodulator pairs with similar effector gene profiles predominantly were “matched” pattern, whereas neurotransmitter pairs with similar effector gene profiles mainly were a “convergent” pattern (Figure 3E). The former suggests the generation of similar neuromodulator pairs shared the specific TF program (“stereotyped programming”), and the latter suggests the generation of similar neurotransmitter pairs could be generated by different TF programs (“flexible programming”). These different programming strategies may account for the fact that across species, neuromodulator types are more conserved, whereas neurotransmitter types are much diverse and variable (La Manno, Gyllborg et al. 2016, Saunders, Macosko et al. 2018, Tiklova, Bjorklund et al. 2019, Poulin, Gaertner et al. 2020).

7. p14, lines 287-297, on the other hand… this part seems redundant with the previous parts and, to me, was difficult to read. Please clarify.

Sorry for the confusion. We re-write this part in the revision (p15, lines 309-314).

On the other hand, in glutamatergic/GABAergic neuronal classification, we also found that 13 pairs of sister clusters with similar TF profiles were separated in the effector gene-based classification. In other words, different from matched and convergent pairs mentioned above (Figure 3B-D), these 13 paired clusters exhibited different effector gene profiles but similar TFs, here terms as “divergent” pattern (Figure 3—figure supplement 4A). Also, the population-level statistical analysis identified divergent pairs (n = 15, Figure 3—figure supplement 4B) that were largely overlapped with those identified by hierarchical sister analysis (n = 10, Figure 3B). Neurons in each of these 10 paired neuronal clusters could be from the same (n=6) or different (n=4) brain regions (Figure 3—figure supplement 4A).

8. p14-15. I admire their efforts to make many BACs. They mentioned this may support "divergent pattern"(p17, line 358), but the reason why they thought so is very weak. They need to show relevance between morphology and effector type somehow experimentally.9. p16, line 340. They did not describe how they picked up 6 TFs.

Thanks for your suggestions. We have clarified this issue in the revision (p16, lines 332-338 and p18, line 391-395).

Neuronal morphology and effector gene profile are two critical criteria of neuron diversity classification (Sugino, Clark et al. 2019, Peng, Xie et al. 2021). Also, many previous studies have provided the apparent links between effector genes and neuron morphology (Kristin L. Whitford 2002, Marcette, Chen et al. 2014, Delandre, Amikura et al. 2016, Noblett, Wu et al. 2019, Peng, Xie et al. 2021). Thus, after we found three patterns (“matched”, “convergent”, and “divergent”), we then wondered if similar patterns present between morphological subtypes and TFs. Indeed, we found in the optic tectum that single TFs could mark multiple morphological subtypes, and single morphological subtype could be marked by multiple TFs, which was in agreement with “divergent” and “convergent” patterns, respectively, considering that divergent pattern indicated the link of same TFs to different neuron phenotypes while convergent pattern indicated the link of different TFs to similar neuron phenotypes.

10. Figure3—figure supplement 2D and 2F. Figure3—figure supplement 4B. I could not tell how to read these data. In addition, in all cases, the effector-based distances are small (between 0-0.1) and TF-based distances are large (between 0.1-0.4). I don't see much difference between these populations.

Thanks for the question. We have clarified this issue in the revision (p17-18, line 372-375).

We selected 6 TFs (*en2b, foxb1a, zic1, bhlhe22, zbtb18*, and *irx1a*) for further analysis based on two criterions: 1. These TFs are highly expressed and specific to individual tectal glutamatergic clusters based on single-cell RNAseq analysis; 2. Their BAC plasmids could reliably mark particular morphological subclasses (at least in four animals).

Reviewer #2:I have to say the authors did not address my concerns at all.My first concern was about the validity of "matched", "convergent" and "divergent" classifications from hierarchical clusterings that are based on effector genes and TF genes in Figures 3 and 5. As an answer, the authors calculated Jaccard similarity indices for clusterings using all genes in Figure 1 and Figure 2. These added analyses were completely unrelated to the clustering of "matched", "convergent" and "divergent." Therefore, I have to say the authors did not address the issue.

Sorry for the confusion.

We first performed the Jaccard similarity index to address the concerns raised by reviewers in the first review regarding the robustness of 68 transcriptome-based clusters.

Then, to verify "matched", "convergent", and "divergent" patterns, we have enlisted the following analyses:

1. Considering that matched, convergent, and divergent patterns are derived from hierarchical clustering or distance measure, which could be sensitive to the tree structure. We have performed population-level statistical analysis in the last revision based on TF-based and effector-based distance measures to re-identify "matched", "convergent", and "divergent" cluster pairs. These re-identified pairs mostly recapitulate those identified using hierarchical clustering (Figure 3B), indicating the robustness of three patterns.

2. Furthermore, we performed the population-level statistical analysis by sub-sampling of genes (80% of total either TF or effector genes) and average statistics over 20 times in this revision to re-identify "matched", "convergent", and "divergent" cluster pairs in Figure 3—figure supplemental 4C-E. These re-identified pairs completely recapitulated those identified using the population-level statistical analysis based on total genes (Figure 3-supplemental 4E), again indicating the robustness of three patterns.

My second concern was about the validity of their claim about the differential expression of RBPs in "divergent cell population". They claim that the RBPs have more divergent expressions in "divergent cell pairs". However, it is hard to interpret the statistics presented in Figure 5E. The indication of the index [(number of genes that are solely divergent in group A) / (number of all genes that are divergent in group A)] is unclear.

Sorry for the confusion. In the revision, we have updated the result in p20, lines 420-427.

The number of RBPs that were specific to divergent pattern (n=52 from 10 pairs) was much more than those specific to convergent pattern (n=10 from 7 pairs, Figure 5—figure supplementary 1E-F). Thus, in terms of the number of pattern-specific RBPs per pair, RBPs showed the preferential expressions in divergent pairs compared to convergent ones (Figure 5D). Note that since there was only one pair of matched pattern, we did not include it in this analysis.

Moreover, this analysis does not indicate any causality between the divergence of RBPs and the divergence of effector genes. In principle, the more heterologous the population is, we would see more divergence of RBPs and effector genes. These factors are not independent of each other. I was expecting to see a more careful statistical approach that takes these considerations into account. Therefore, I have to say the authors did not address the issue.

Thanks for your suggestion. In the revision, we have updated the result in p20-21, lines 433-442.

We have enlisted a new analysis on the probability of effector gene which are the binding sites of pattern-specific RBPs using oRNAment database (http://rnabiology.ircm.qc.ca/oRNAment/) to address a potential causality between the divergence of RBPs and the divergence of effector genes (Benoit Bouvrette, Bovaird et al. 2020). GO analysis showed that pattern-specific RBPs targeted various molecular categories including effector genes, TFs, mRNA processing, metabolism (Figure 5—figure supplement 1H-J, Figure 5-Source Data 4). More importantly, significantly higher proportions of effector genes (in all targeted genes) are bound by divergent pattern-specific RBPs than by other two pattern-specific RBPs (Figure 5E). These results suggested the causality between RBPs and effector gene profiles in divergent pairs.

Unfortunately, due to the low gene coverage of 10x Genome-based scRNAseq and the limited number of targeted genes available in the database, we were unable to directly analyze the causality between the divergence of RBPs and the divergence of effector genes at the resolution of single divergent pairs. It is necessary in the future to directly examine the binding of RBPs and effector marker genes in divergent pairs.

Reviewer #3:In the revised manuscript, the authors mainly added three additional control analyses that increased the credibility of the conclusion, including:1) Robustness analysis of the clustering result using Jaccard similarity of subsampled data.2) Analysis of the matched/convergent/divergent patterns of the glutamatergic/GABAergic neurons based on relaxed criteria that include not only the terminal sister clusters, but also non-sibling clusters with relatively close relationships.3) Control analysis to show the enrichment of differential RBP expression in divergent patterns.The control analyses largely addressed my concerns to a large extent, although they made the dramatic contrast between the regulatory logic of neurotransmitter and neuromodulator data weaker. The identified convergent/divergent events are now more convincing and allow for further exploration of the phenotype in the future.However, I cannot help but noticed many more changes have been made without being pointed out in the rebuttal letter, including (1) exclusion of the data from the cholinergic neurons altogether,

Thanks for the question.

We did remove the data from the cholinergic neurons due to the following reasons:

1. Recently, we have noticed that the newly-generated transgenic line Tg(*chata*:EGFP) showed some levels of variability in marking cholinergic neurons at the whole-brain scale. Currently, we continue verifying this line and screening for new stable lines.

2. Also, considering that whole-brain scRNAseq data also showed the preferential co-expression of choline and glutamate (Figure 1-Source Data 2), the exclusion of the data from cholinergic neurons does not influence any conclusion in the paper, we then decided to exclude this data set from the revised version to ensure the quality of scRNA data in this Resource paper. Meanwhile, we described the above results in the revision (p10-11, line 218-220).

Sorry for this ignorance in the last response letter and we hope the reviewers can accept this request.

and (2) slightly different analysis result with seemingly identical input data (e.g. comparing Figure 3—figure supplement 2 in the new version vs. Figure 3 in the original manuscript). The authors should explain why such changes were made and ensure there is no selective report of the data.

Sorry for the confusion. In the revision, we have introduced new analysis in p11, lines 240-242.

We did include three new matched sister pairs (30-23, 4-32, 4-22) in the last revision because they were the pairs with the lowest 10% of distance measures using population-level statistical analysis, even though they were not immediate sister clusters at hierarchical termini (Figure 3-figure supplementary 2C and 4A).

However, in this revision, we identify matched pairs in hierarchical clustering by applying more rigorous criteria: 1. They are terminus sister clusters; 2. They were identified as matching modes based on R package “TreeDist”. The result showed that Cluster 9-61 was the only matched pair (Figure 3-figure supplementary 1B and 2B).

Finally, some suggestions/questions to help achieve a better presentation of the data.1) The current "population-level" analysis is rather just a relaxed definition of the terminal sister clusters, but there are many other metrics to quantify the similarity between the TF-based and effector-based clustering result over the entire dataset. For hierarchical clustering this can be editing distance between the two trees, or more generally, similarity between organization of TF and effector gene distribution across all the clusters. This will not identify specific matched, convergent or divergent patterns, but would be an unbiased measurement of how TF and effector landscape agree with each other.

Thank you for your suggestions. We introduce R package “treeDist” to calculate tree distance in the revision (p11-12, line 241-242 and Materials and methods p41, line 794-801).

In this revision, we first tried “editing distance”. Unfortunately, since there are a total of 36 nodes (for 39 neuronal clusters) in both TF-based and effector gene-based hierarchical clusters, we were unable to run “editing distance” on such big trees successfully. Alternatively, we quantified the similarity between TF-based and effector gene-based hierarchical clusters by tree distance using R package “TreeDist”.

Remarkably, the analysis identified only one matching node (Cluster 9-61) after comparing TF-based and effector gene-based clusters (Figure 3—figure supplementary 1B). This paired cluster (Cluster 9-61) was consistent with the matched pair identified by hierarchical clustering (Figure 3—figure supplementary 2B) and the population-level statistical analysis (Figure 3—figure supplementary 2D).

Moreover, the tree distance between TF-based and effector gene-based clusters was 0.71 (Figure 3—figure supplementary 1B, p11-12, line 241-242). Meanwhile, we quantified the distance of two similar trees that was exemplified by TF-based or effector gene-based clusters with or without subsampling of 80% genes, and the result showed that the tree distances of TF-based or effector gene-based clusters before and after subsampling were only 0.20 (TF-based) and 0.14 (effector gene-based), respectively (Figure 3—figure supplementary 4C-D).

All above results indicated the overall distinction of the landscape of TF-based and effector gene-based clusters.

2) Figure 3: similar color regime was used to represent forebrain/OT/sub OT and matched/convergent/divergent patterns. This is very confusing. I would suggest weakening brain region visualization (e.g. changing it to a less pronounced representation or getting rid of it altogether -see comment #3) and focus on the regulatory logic. It is also recommended to color the pattern names in 3B accordingly.

Thanks for the suggestion. We have updated the color regime in the revision as follows:

1. We removed the brain region visualization and focused on the regulatory logic in Figure 3C-D, and 3F-G.

2. We have colored the pattern names in Figure 3B, 3C-D, and 3F-G and Figure 5B.

3) Figure 3C: The number of matched patterns on the Venn diagram does not match the number shown in Figure 3F. Also, Venn diagram is not the best way to show that the conclusion in sister cluster- and population-statistics are similar. [Minor correction: the number of selection should be Combination C(39, 2) rather than Permutation A(39, 2)].

Thank you for your suggestions.

1. We have updated the Venn diagram in Figure 3B in the revision.

2. We have updated the number of selection using the combination C(39, 2) in Figure 3A in the revision.

4) Throughout the manuscript, the brain region identities are shown next to the clusters in every figure, yet very little conclusion was made about the relationship between transcriptome and brain region identity. The authors are advised to summarize the major findings of data feature related to brain region identities (no correlation can also be a finding), and simplify the color use of brain region when the color does not convey meaningful information.

Thanks for your suggestions. We have discussed the significance of brain region identity in the revision (p24, line 513-527).

Regional identity is an essential factor for classifying brain cells. Our results showed that neurons, qRG, and neuronal progenitors exhibited prominent regional characteristics (Figure 1C). In the hierarchical clustering and population-level statistical analysis, similar pair clusters from the same region or different regions occur in nearly equal probabilities (Figure 3—figure supplement 2A). Because neuronal phenotypes from the same region could exhibit common regional identities, it was surprising to observe that such a higher proportion of sister subclasses with similar transcriptomic phenotypes could arise from two different brain regions. Putting this finding into the context of neuron type evolution raises the possibility that different brain regions independently give rise to similar neurotransmitter phenotypes through different TF programs. Alternatively, as brain regions are functionally diversified during the evolution, these highly similar neuronal clusters of different brain regions possibly derive from ancient building blocks, which become divergent through the evolutionary acquisition of different combinatorial TF codes.

Reference

Benoit Bouvrette, L. P., S. Bovaird, M. Blanchette and E. Lecuyer (2020). "oRNAment: a database of putative RNA binding protein target sites in the transcriptomes of model species." Nucleic Acids Res **48**(D1): D166-D173.

Delandre, C., R. Amikura and A. W. Moore (2016). "Microtubule nucleation and organization in dendrites." Cell Cycle **15**(13): 1685-1692.

Hodge, R. D., T. E. Bakken, J. A. Miller, K. A. Smith, E. R. Barkan, L. T. Graybuck, J. L. Close, B. Long, N. Johansen, O. Penn, Z. Yao, J. Eggermont, T. Hollt, B. P. Levi, S. I. Shehata, B. Aevermann, A. Beller, D. Bertagnolli, K. Brouner, T. Casper, C. Cobbs, R. Dalley, N. Dee, S. L. Ding, R. G. Ellenbogen, O. Fong, E. Garren, J. Goldy, R. P. Gwinn, D. Hirschstein, C. D. Keene, M. Keshk, A. L. Ko, K. Lathia, A. Mahfouz, Z. Maltzer, M. McGraw, T. N. Nguyen, J. Nyhus, J. G. Ojemann, A. Oldre, S. Parry, S. Reynolds, C. Rimorin, N. V. Shapovalova, S. Somasundaram, A. Szafer, E. R. Thomsen, M. Tieu, G. Quon, R. H. Scheuermann, R. Yuste, S. M. Sunkin, B. Lelieveldt, D. Feng, L. Ng, A. Bernard, M. Hawrylycz, J. W. Phillips, B. Tasic, H. Zeng, A. R. Jones, C. Koch and E. S. Lein (2019). "Conserved cell types with divergent features in human versus mouse cortex." Nature **573**(7772): 61-68.

Kristin L. Whitford, V. r. M., Elke Stein, Corey S. Goodman, Marc Tessier-Lavigne, Alain Che´ dotal, and Anirvan Ghosh (2002). "Regulation of Cortical Dendrite Development by Slit-Robo Interactions." Neuron **33**: 47-61.

La Manno, G., D. Gyllborg, S. Codeluppi, K. Nishimura, C. Salto, A. Zeisel, L. E. Borm, S. R. W. Stott, E. M. Toledo, J. C. Villaescusa, P. Lonnerberg, J. Ryge, R. A. Barker, E. Arenas and S. Linnarsson (2016). "Molecular Diversity of Midbrain Development in Mouse, Human, and Stem Cells." Cell **167**(2): 566-580 e519.

Marcette, J. D., J. J. Chen and M. L. Nonet (2014). "The *Caenorhabditis elegans* microtubule minus-end binding homolog PTRN-1 stabilizes synapses and neurites." *ELife*
**3**: e01637.

Noblett, N., Z. Wu, Z. H. Ding, S. Park, T. Roenspies, S. Flibotte, A. D. Chisholm, Y. Jin and A. Colavita (2019). "DIP-2 suppresses ectopic neurite sprouting and axonal regeneration in mature neurons." J Cell Biol **218**(1): 125-133.

Paul, A., M. Crow, R. Raudales, M. He, J. Gillis and Z. J. Huang (2017). "Transcriptional Architecture of Synaptic Communication Delineates GABAergic Neuron Identity." Cell **171**(3): 522-539 e520.

Peng, H., P. Xie, L. Liu, X. Kuang, Y. Wang, L. Qu, H. Gong, S. Jiang, A. Li, Z. Ruan, L. Ding, Z. Yao, C. Chen, M. Chen, T. L. Daigle, R. Dalley, Z. Ding, Y. Duan, A. Feiner, P. He, C. Hill, K. E. Hirokawa, G. Hong, L. Huang, S. Kebede, H. C. Kuo, R. Larsen, P. Lesnar, L. Li, Q. Li, X. Li, Y. Li, Y. Li, A. Liu, D. Lu, S. Mok, L. Ng, T. N. Nguyen, Q. Ouyang, J. Pan, E. Shen, Y. Song, S. M. Sunkin, B. Tasic, M. B. Veldman, W. Wakeman, W. Wan, P. Wang, Q. Wang, T. Wang, Y. Wang, F. Xiong, W. Xiong, W. Xu, M. Ye, L. Yin, Y. Yu, J. Yuan, J. Yuan, Z. Yun, S. Zeng, S. Zhang, S. Zhao, Z. Zhao, Z. Zhou, Z. J. Huang, L. Esposito, M. J. Hawrylycz, S. A. Sorensen, X. W. Yang, Y. Zheng, Z. Gu, W. Xie, C. Koch, Q. Luo, J. A. Harris, Y. Wang and H. Zeng (2021). "Morphological diversity of single neurons in molecularly defined cell types." Nature **598**(7879): 174-181.

Poulin, J. F., Z. Gaertner, O. A. Moreno-Ramos and R. Awatramani (2020). "Classification of Midbrain Dopamine Neurons Using Single-Cell Gene Expression Profiling Approaches." Trends Neurosci **43**(3): 155-169.

Saunders, A., E. Z. Macosko, A. Wysoker, M. Goldman, F. M. Krienen, H. de Rivera, E. Bien, M. Baum, L. Bortolin, S. Wang, A. Goeva, J. Nemesh, N. Kamitaki, S. Brumbaugh, D. Kulp and S. A. McCarroll (2018). "Molecular Diversity and Specializations among the Cells of the Adult Mouse Brain." Cell **174**(4): 1015-1030 e1016.

Sugino, K., E. Clark, A. Schulmann, Y. Shima, L. Wang, D. L. Hunt, B. M. Hooks, D. Trankner, J. Chandrashekar, S. Picard, A. L. Lemire, N. Spruston, A. W. Hantman and S. B. Nelson (2019). "Mapping the transcriptional diversity of genetically and anatomically defined cell populations in the mouse brain." *ELife*
**8**.

Tiklova, K., A. K. Bjorklund, L. Lahti, A. Fiorenzano, S. Nolbrant, L. Gillberg, N. Volakakis, C. Yokota, M. M. Hilscher, T. Hauling, F. Holmstrom, E. Joodmardi, M. Nilsson, M. Parmar and T. Perlmann (2019). "Single-cell RNA sequencing reveals midbrain dopamine neuron diversity emerging during mouse brain development." Nat Commun **10**(1): 581.

[Editors' note: further revisions were suggested prior to acceptance, as described below.]

1) Although for neurotransmitter-type, "sister cluster analysis and the population-level statistical analysis " are carried out. However, for neuromodulator-type, only sister cluster analysis was done. It is fair to perform population-level analysis for neuromodulator-type.

Thanks for the suggestion. We have performed the population-level statistical analysis on neuromodulator-type neurons in this revision (line 305-318; Figure 3—figure supplementary 3E-G).

In this revision, we first updated the analysis based on hierarchical sister clusters by applying more rigorous criterions: 1. They are terminus sister clusters; 2. They were identified as matching nodes based on R package “TreeDist”. The analysis showed that 5 out 8 sister neuromodulator clusters with matched pattern (Figure 3—figure supplementary 3D). In the previous analysis, 7 out of 8 sister neuromodulator clusters were identified as matched pattern, but 2 pairs of similar clusters (4/14; 6/21) were not immediate sister clusters at the hierarchical termini (Figure 3—figure supplementary 3D′).

Meanwhile, we performed the population-level statistical analysis using all genes or subsampling of 80% genes to identify similar neuromodulator pair clusters. We identified 9 similar neuromodulator pair clusters, 8 with matched pattern and 1 with convergent pattern (Figure 3—figure supplementary 3F).

Thus, we intersected hierarchical sister cluster analysis and population-level statistical analysis, and identified 6 pairs of similar neuromodulator clusters, 5 with matched pattern and 1 with convergent pattern (Figure 3—figure supplementary 3G-I).

2) Line 265: largely recapitulated. I do not agree with this. Rather, explain why the results were not consistent.

Thanks for the suggestion. We re-write this part and explain the results in the revision (Line 265-279) as followings:

“The population-level analysis identified 19 pairs of effector gene-based similar pair clusters, 5 with matched pattern and 14 with convergent pattern (Figure 3—figure supplement 2D-E). Overall, similar pair clusters with either matched or convergent pattern identified by the population-level analysis showed an overlapping but distinct pattern with those identified by the hierarchical sister cluster analysis (Figure 3B). This discrepancy was likely due to the following facts: 1. In the population-level statistical analysis, we arbitrarily set the lowest threshold as a criterion to identify similar pair clusters. Thus, the levels of this threshold could influence the production of similar pair clusters. 2. In population-level statistical analysis, each cluster could use for multiple times, whereas in hierarchical sister cluster analysis, once a cluster was selected as a pair with another cluster, it could not be re-used again. To overcome this discrepancy, we intersected the results from hierarchical sister cluster analysis and population-level statistical analysis, and identified 8 pairs of effector gene-based glutamatergic/GABAergic similar pair clusters, 1 with matched pattern and 7 with convergent pattern (Figure 3B)”.

3) Line 226: The TF Regulatory Landscape in Whole-brain Neuronal PhenotypesThis section is a bit too long. The section should be separated, for example, for neurotransmitter-type and neuromodulator-type.

Thanks for the suggestion.

We have separated this section into three parts: 1. The TF Regulatory Landscape of Whole-brain Glutamatergic/GABAergic Neuron Clusters (Line 226-227); 2. The TF Regulatory Landscape of Neuromodulatory Neuron Clusters (Line 304); 3. Different Neuron Clusters Exhibit Similar TF Profiles (Line 332).

4) Line 395: results supported "divergent" and "convergent" patterns, because divergent pattern indicated the link of same TFs to different neuron phenotypes, while convergent pattern indicated the link of different TFs to similar neuron phenotypes.This is too much speculation. The relationship between the morphology and effector gene profiles are unclear. I think it is fair to say that morphology does not directly correlate with TF profiles.

Thanks for the suggestion.

We have updated our statement in the revision as “This observation could be inferred from “convergent pattern” (same TFs were expressed in different effector-based subtypes) and “divergent pattern” (different TFs were expressed in similar effector-based subtypes), respectively. However, we could not exclude unknown indirect regulations of morphological subtypes by TFs” (Line 427-431).

Reviewer #2:I can say the authors addressed my concerns on statistical analyses after this round of revision, shrugging off unnecessary parts and exposing the most interesting part of this study. In addition, they included a new analysis in Figure 5 which indicates potential causality between the differential expression of RBPs and differential expression of effector genes.There are still concerns about the validity of classification (lower 10% of gene distance compared to shuffled pairs, how do you justify the threshold?). However, this justification will not be so straightforward because it will be prone to the intrinsic statistics of TF expression, effector gene expression, and the sensitivity of the sequencing technique. It is obvious this is the best authors can do. I advise authors to acknowledge such limitations in the discussion.

Thanks for the suggestion. We have included the discussion on this limitation in the revision in both Result and Materials and methods.

In Results (line 270-279): “In the population-level analysis, we arbitrarily set the lowest threshold as a criterion to identify similar pair clusters. Thus, the levels of this threshold could influence the production of similar pair clusters.”

In Materials and methods (line 828-829): “This analysis is subject to the intrinsic statistics of TF and effector gene expression, as well as the sensitivity of scRNAseq method”.

In addition, the reason why we applied the lowest 10% threshold is this intrinsic difference in TF-based (0.1-0.4) and effector-gene-based (0-0.1) distances between clusters (Figure 3—figure supplementary 2D-E). One possible explanation is due to the difference in the numbers of TF genes (less, n = 283) and effector genes (more, n = 1099). However, considering that subsampling of genes has little influence on identifying matched, convergent, and divergent types (Figure 3-supplemental 2H), it ruled out the contribution of the difference in the number of TFs and effector genes to the difference in distance measures. Thus, this distance difference more likely resulted from the hierarchical regulation of effector gene expression by TFs.

Reviewer #3:Unfortunately, the authors misunderstood my request for adding Tree Distance analysis. An important claim of the paper was that glutamatergic/GABAergic clusters show different TF and terminal profile patterns ("convergent/divergent"), whereas neuromodulator type clusters predominantly expressed the same TF profiles ("matched"). However, supplementary analysis was only shown for the glutamatergic/GABAergic types but not the modulator types, and the main figure that supports this conclusion (Figure 3E) still uses sister clusters from hierarchical clustering for classification of neuromodulator types, which I found to be an unfair comparison.

Thanks for the great suggestion. We have performed the population-level analysis on neuromodulator-type neurons to strengthen the conclusion in this revision (line 312-318; Figure 3—figure supplementary 3E-G). Meanwhile, we also updated similar neuromodulator pairs using the hierarchical sister cluster analysis by applying more rigorous criterions (line 306-310; Figure 3—figure supplementary 3C-D).

We first updated the analysis based on hierarchical sister clusters by applying more rigorous criterions: 1. They are terminus sister clusters; 2. They were identified as matching nodes based on R package “TreeDist”. The analysis showed that 5 out 8 sister neuromodulator clusters with matched pattern (Figure 3—figure supplementary 3C-D). In the previous analysis, 7 out of 8 sister neuromodulator clusters were identified as matched pattern, but 2 pairs of similar clusters (4/14; 6/21) were not immediate sister clusters at the hierarchical termini (Figure 3—figure supplementary 3D′).

Meanwhile, we performed the population-level statistical analysis using all genes or subsampling of 80% genes to identify similar neuromodulator pairs. We identified 9 similar neuromodulator clusters, 8 with matched pattern and 1 with convergent pattern (Figure 3—figure supplementary 3 E-F).

Thus, we intersected hierarchical sister cluster analysis and population-level statistical analysis, and identified 6 pairs of similar neuromodulator clusters, 5 with matched pattern and 1 with convergent pattern (line 315-318; Figure 3—figure supplementary 3G-I).

Also, the gene subsampling test of the robustness of matched/divergent/convergent pairs was only performed on similar pairs defined by population-statistics. It was not clear to me why the authors still adhered to the "sister cluster" definition (or an INTERSECTION of this strategy with the population-statistics), for all subsequent conclusions.

Thanks for the question.

We have addressed this concern in the revised manuscript (Line 267-279)

“Overall, similar neuron clusters with either matched or convergent pattern identified by the population-level analysis showed an overlapping but distinct pattern with those identified by the hierarchical sister cluster analysis (Figure 3B). This discrepancy was likely due to the following facts: 1. In the population-level analysis, we arbitrarily set the lowest threshold as a criterion to identify similar pair clusters. Thus, the levels of this threshold could influence the production of similar pair clusters. 2. In population-level statistical analysis, each cluster could use for multiple times, whereas in hierarchical sister cluster analysis, once a cluster was selected as a pair with another cluster, it could not be re-used again. To overcome this discrepancy, we intersected the results from hierarchical sister cluster analysis and population-level statistical analysis, and identified 8 pairs of similar glutamatergic/GABAergic clusters, 1 with matched pattern and 7 with convergent pattern (Figure 3B)”.

We think it is better to keep all these analyses in this manuscript to strengthen our conclusions. We also greatly appreciate all reviewers’ suggestions to make the conclusion more solid.

I suggested tree distance as a global measurement of whether the glutamatergic/GABAergic types and modulator types truly have different regulatory logic at the population level, rather only at the terminal sisters.

Thanks for the suggestion.

In the revision, we discussed different regulatory logic at the population level between neuromodulator-type and neurotransmitter-type neurons in Line 323-332:

“we have performed this global tree measurement using the R package “TreeDist”, and found that the distance between TF-based and effector-based hierarchical tree of glutamatergic/GABAergic neurons (Tree distance = 0.71) was higher than that of neuromodulator neurons (Tree distance = 0.38), suggesting distinct TF regulatory logic of effector-based phenotypes between neurotransmitter-type and modulator-type at the global level. The smaller distance for neuromodulator neurons was consistent with our conclusion that neuromodulator type clusters predominantly expressed the same TF profiles ("matched")”.